# Wind resource assessment Offshore Fujian using 30-year wind estimates

**Wenfei Xue[1], Rongfu Li[1], Weidong Ji[1], Endi Zhai[1], Hongying Yang[1], Qiaozhen Ning[1], Xiaokai Hu[2], Guanghong Liao [3]***

**1** Goldwind Science & Technology Co., Ltd, Urumqi, China, **2** NARI Technology Co. Ltd, Nanjing, China, **3** Key Laboratory of Marine Hazards Forecasting, Ministry of Natural Resources, Hohai University, Nanjing, China

* liaogh@hhu.edu.cn

## Abstract

China has set ambitious goals for the development of offshore wind energy to meet the increasing energy needs of coastal provinces. The initial phase of offshore wind energy development involves evaluating the wind resource and identifying the most promising locations for wind farms. It is crucial to assess the characteristics and potential of wind energy beforehand. This study conducts a comprehensive assessment of offshore wind resource near Fujian China. Wind measurement devices were deployed at XiaPu and PingTan to collect wind profile data and meteorological conditions for one year. Various wind characteristics, including average wind speed, frequency of wind direction, wind shears and turbulence intensity were analyzed. An adaptive Measure-Correlate-Predict methodology was utilized to estimate wind conditions over 30-years span. Measured Wind energy density values range from 3082.63 and 11753.52 kWh/m²/year. The peak daily average wind speeds are prevailing between 12 a.m. and 11 p.m with lower turbulence intensity and higher wind shear exponent, such condition is suitable for development of wind power. The variation in the wind shear exponent, and wind speed changes with the seasons. The 90th percentile of turbulence intensity was found to be below the standard set for IEC Class A+ The extreme wind speed associated with a 50-year return period was 38.0m/s at a height of 100m, leading to the recommendation of wind turbine class II. However, taking into account the ambient turbulence intensity, it might be advisable to upgrade the turbine class to IEC Class A⁺.

## 1. Introduction

As the global economy expands, energy demand worldwide is increasing. The need to develop renewable energy solutions is growing, driven by climate change impacts and the decreasing supply of fossil fuel resources. The 2023 Renewable Energy Annual Market Report from the International Energy Agency indicates that global

**Data availability statement:** All data files are available from the Zenodo database (accession number: 10.5281/zenodo.15254530).

**Funding:** National Key Research and Development Program of China (2022YFB4201400 and 2024YFB4207201).

renewable energy capacity increased by 50% in 2023, reaching a total of 510 GW. This growth rate is the highest seen in the past 30 years. Additionally, the report predicts an even faster increase in renewable energy capacity over the next five years. Among various renewable energy sources, wind power stands out as the fastest growing option, gaining global recognition due to advancements in technology that improve its efficiency. Notably, offshore wind energy provides higher energy density, fewer spatial constraints, lower risk of civil disputes, and greater efficiency than onshore wind energy, making it a more viable alternative to fossil fuels, which contribute to global warming. For example, 1 GW of offshore wind energy can reduce over 3.5 MT of CO2 emissions [1]. By 2050, the renewable energy is projected to account for 86% of global electricity, up from the current 29% [2]. China is making significant strides in renewable energy, emerging as a key player in global climate change initiatives. In 2023, China installed 305 GW of new renewable energy capacity, with wind and photovoltaic (PV) power accounting for 292.78 GW, which meets 33% of its total electricity consumption.

While wind energy is widely recognized as one of the most sustainable forms of renewable energy. Regions such as South America, Asia, Africa, and Europe, have conducted research to assess offshore wind resources. Marcolino et al. analyzed the hourly availability of offshore wind and solar energy resources along Brazil's coastline [3]. Chancham et al. assessed the offshore wind resource in the Gulf of Thailand using data from numerical models and reanalysis [4]. Four sets of reanalysis wind data were investigated to evaluate their effectiveness in estimating offshore wind energy, using reference data from an offshore meteorological mast in the southwestern peninsula [5]. Ohunakin et al.utilized long-term observational data to analyze the technical and economic feasibility of offshore wind power in the Gulf of Guinea [6]. Annual Energy Productions (AEPs) for 2.3 MW wind turbines in the Thracian Sea were calculated using wind data from the Copernicus Marine Environmental Service (CMEMS) scatterometer from 2011 to 2019 [7]. Takvor et al. assessed the potential and variability of offshore wind energy and its variability in the Mediterranean across different time scales [8]. Ferrari et al. identified optimal offshore sites for harnessing Mediterranean wind and wave energy through simulations conducted with WRF and WWIII [9].

Considering the availability of substantial reserves, a comprehensive analysis of wind patterns and their potential is essential to evaluate the feasibility of installing offshore wind turbines [10–11]. Kim et al. established specific guidelines for identifying optimal offshore locations in Korea's southwestern waters noting limited social and environmental conflicts and promising economic viability [12]. Wu et al. introduced a framework to aid in the decision-making process for offshore wind farms site selection [13]. Additionally, Sørensen and Larsen presented a rapid and effective optimization model for evaluating wind resources and associated costs during the development of large-scale offshore wind farms [14].

With the swift increase in demand for renewable energy, China has made considerable advances in wind energy development since 2006. Onshore wind power constitutes predominant segment of the wind energy sector, with installations

primarily located in northern and northwestern China. However, the existing transmission infrastructure is inadequate for the efficient transfer of substantial wind power to the eastern and southeastern coastal provinces. Conversely, offshore wind energy presents an optimal solution to supplement thermal and hydroelectric power, thereby addressing energy consumption demands in China's coastal regions. So, the Chinese government has focused on the possibilities presented by offshore wind power in recent years. China is one of the largest countries in terms of installed offshore wind power capacity globally. By the end of 2024, the cumulative installed capacity exceeded 30 GW, accounting for more than 40% of the global total, and mainly concentrated in coastal provinces such as Jiangsu, Guangdong, Fujian, and Zhejiang.

The initial phase of offshore wind energy development involves evaluating the wind resource and pinpointing the most promising locations for wind farms, and it is essential to assess the characteristics and potential of wind energy beforehand. Numerous studies have assessed the offshore wind energy potential in China's coastal waters utilizing observational data, reanalysis datasets, and model outputs [15–22]. Qin et al. conducted a study on offshore wind energy along the Chinese coasts at a 100-meter hub height utilizing the MM5 model [23]. Their findings indicated that the coastal regions of southeastern China possess the most abundant offshore wind resources compared to other parts of the Chinese coastline. The available offshore wind energy potential within China's exclusive economic zone seas has been evaluated, considering technical, spatial, and economic constraints, through GIS [24]. Their findings suggest that shallow waters along China's eastern coast are favorable for the development of offshore wind energy, and the economic potential of offshore wind power could meet 56%, 46%, and 42% of the electricity demands of coastal regions by 2010, 2020, and 2030, respectively. Zheng et al. calculated the China sea wind energy density using CCMP wind data spanning from 1988 to 2009 [15]. They find that the majority of the China Sea contains substantial offshore wind energy resources, with wind energy densities exceeding 150 W/m², and wind energy storage surpassing $2 \times 10^3$ kWh/m². The most prolific region is located in the northern South China Sea, followed by the southern South China Sea and the East China Sea, the Yellow Sea and Bohai Sea possess relatively lower wind energy resources. A climatological review of wind power resources at 100-m height over the Bohai Sea and the Yellow Sea was conducted utilizing high-resolution wind speed data reconstructed from reanalysis and regional models [18]. Based on daily QuikSCAT wind field data spanning from 2000 to 2008, Jiang et al. evaluated the spatial-temporal variability of China's offshore wind resources [25]. Their analysis revealed that the total 10-meter offshore wind energy potential approximates 660 GW. According to the NCEP-CFSR reanalysis data from 1991 and 2010, Wang et al. determined that the annual mean wind power density exceeds 200 W/m², with over 4,000 cumulative hours of wind speeds surpassing 6 m/s at a height of 70 meters along China's coastal regions [26]. Offshore wind resources in Shenzhen's coastal areas were studied using observed data [20]. The average wind speed at a height of 2.5 meters ranges from 3.1 m/s and 4.1 m/s. Analysis of offshore wind in the South China Sea shows that regions with the highest wind intensity are primarily found in the Luzon and Taiwan Straits [22].

The literature consistently indicates that China's offshore wind energy resources are substantial and can significantly augment national energy production. However, several challenges persist in the assessment of offshore wind energy. Firstly, most evaluations, relying on satellite, reanalysis or model data, are conducted at sea surface wind heights of 10 meters or below hub height levels. Observations from meteorological masts, especially at hub height, are sparse in the offshore areas, and often have limitations, such as time inconsistencies and spatial and temporal limitations because of the construction of meteorological masts is expensive and time consuming. Thus, it is lack of refine and high-accuracy analysis, especially in the planning and design of regional wind farms. Secondly, the datasets employed for wind energy potential assessments are typically low temporospatial resolution, failing to capture wind turbulence intensity, wind shears and finer details of offshore wind energy. Consequently, accurate wind data are imperative for precise wind power assessment, crucial for site selection and turbine installation. Additionally, prior research predominantly focuses on wind energy assessments, with limited attention given to extreme wind investigations, which are vital for the construction, maintenance, and sustainability assessment of existing wind power installations.

An accurate assessment of the spatiotemporal distribution and characteristics of offshore wind energy is essential for the sustainable development and utilization of China's offshore wind resources. Specifically, wind speeds in the Taiwan Strait, which situated between the provinces of Fujian and Taiwan Island, are intensified due to the narrow tube effect, resulting in a significant peak in maximum wind power density for the region. Consequently, offshore wind farms are widely recognized for their significant potential due to abundant wind resources. Wind speed observation is the basis of wind resource assessment. To ensure high reliability in estimating offshore wind energy potential in the Taiwan Strait, it is crucial to use actual offshore wind measurement data. However, many studies prefer reanalysis data over field measurements due to the significant financial costs and time required for collecting offshore observational data. Meteorologists emphasize the need for at least 30 years of data to assess regional wind climates, yet research utilizing 30-year wind estimates remains limited.

The aim of the present study is to conduct a thorough investigation into the design of offshore wind farms using high frequency wind profile data. Initially, wind characteristics are analyzed using data from two meteorological masts in the Hujian coastal region. A wind resource evaluation follows to determine key design parameters such as wind potential, vertical wind profile, wind power density, and wind energy density. In accordance with IEC 61400 standards, wind turbines are categorized by examining extreme wind speeds and turbulence intensity. To improve the reliability of the analysis, the Measure-Correlate-Predict (MCP) methodology is used to assess wind turbine classification.

The paper is structured as follows: Section 2 provides an overview of the study and observation data. Section 3 details analysis method, including wind speed distribution model, most probable wind speed, air density calculation, wind energy density, wind shears, turbulence intensity and MCP method. Section 4 analyzes the high frequency variability of wind and wind power, along with extreme return winds at a height of 100 meters. Finally, Section 5 summarizes the findings and conclusions.

## 2. Study area and observation data

Fujian's coastal region, particularly the Taiwan Strait, is characterized by a "wind tunnel effect" due to its narrow topography, resulting in some of China's strongest and most stable offshore wind resources. This area serves as a critical hub for China's offshore wind energy development, with over 20% of the nation's installed capacity concentrated here. Fig 1 shows the coastal waters near Hujian, China, indicating the locations of offshore wind observation sites and a grid point for fifth-generation ECMWF atmospheric reanalysis (ERA5) data. Two measurement stations for offshore wind have been set up along the coastline, averaging about 5.0 km from shore, and their geographical coordinates observation

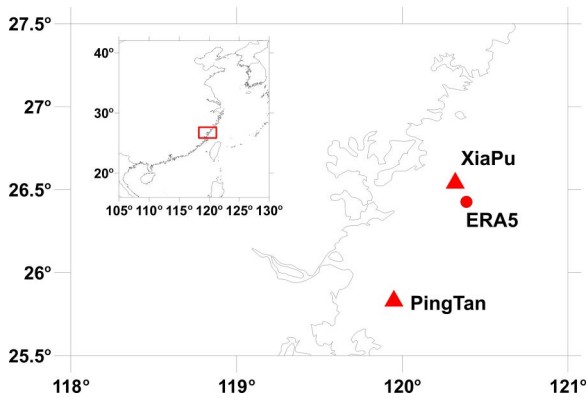

**Fig 1. Location of Fujian wind power farm area in China coastal water.** The offshore met mast and ERA5 reanalysis data station is indicated. The coast line data is derived from public domain map dataset (http://www.naturalearthdata.com/).

periods, as well as observation heights are listed in Table 1. The selection of two stations captures localized wind patterns influenced by monsoons, typhoons, and coastal-terrain interactions, which are underrepresented in existing global data-sets. Hourly 10 m and 100 m wind speed data from ERA5 is evaluated by comparing it with direct wind measurements obtained from 19 wind tower located across the coastal waters of China [27]. They point out that the basic statistical characteristic between ERA5 reanalysis and observed wind speeds demonstrate good consistency. But, the ERA5 tends to underestimate wind speed, particularly at high speeds during extreme conditions [27]. The ERA5 reanalysis data point, located at 26.54°N and 120.32°E, served as the source for retrieving the reference long-term wind data used for climate adjustments.

This research collected wind speed and direction data at heights of 20, 50, 70, 80, 90, and 100 meters above mean sea level (AMSL) using a data logger connected to anemometers and wind vanes. The chosen heights comprehensively consider the balance between characterizing vertical wind profiles, equipment feasibility, and turbine design and energy yield validation. This approach ensures robust data for both theoretical modeling and real-world applications. The Thies First Class sensors employed for these instruments are known for their high measurement accuracy and stability. The specifications of wind sensors are detailed in Table 2. Although the data collection spanned one year, each observation station corresponds to a different year. Data was sampled every second, with the study focusing on analyzing 10-minute averaged wind data.

Before analyzing wind data, it's crucial to assess its quality to achieve reliable results. This evaluation involved checking range, relationships, and trend of the measurement. Data recovery rates at all six heights were 100%, suggesting no sensor failures, misreading, or poor-quality data.

## 3. Analysis methods

### 3.1 Wind speed distribution model

Various two-parameter probability density functions (PDFs), such as the Weibull, lognormal, gamma, and Gumbel distributions, have been used to model wind speed distributions over time [28]. The Weibull distribution is commonly considered the most suitable option, especially for large datasets of recorded wind speeds. However, it has a limitation in accurately

**Table 1. Detailed information of wind speed observation sites, their locations, and observation periods, as well as observation heights.**

| ID | Station Name | Location | | Period (day/month/year) | Height (m) |
|----|--------------|----------|----------|-------------------------|------------|
| | | Longitude | Latitude | | |
| W01 | XiaPu | 120.32°E | 26.54°N | 1/12/2019-31/12/2020 | 100, 90, 80, 70, 50, 20 |
| W02 | PingTan | 119.95°E | 25.83°N | 1/8/2019-31/7/2020 | |

**Table 2. The specification of wind sensors and measurement conditions.**

| Items | Sensors | |
|-------|---------|---|
| | Anemometer | Wind vane |
| Instrument | RM YOUNG05106 | RM YOUNG05106 |
| Measuring range | 0~100m/s | 0~360° |
| Measuring accuracy | ±0.3m/s | ±1° |
| Starting threshold | 1.1 m/s | 1.0m/s |
| Operation temperature | −50°C~60°C | −50°C~60°C |
| Sampling rate | 90 Hz | 90Hz |
| Data averaging | 10 min | 10 min |
| Measurement height | 100m, 90m, 80m, 70m, 50m 20m | 100m, 50m, |

representing the probabilities of zero or very low wind speeds. This shortcoming is generally not significant because the energy produced at low speeds is usually negligible, making it less crucial for evaluating the wind energy potential for commercial turbines.

The Weibull distribution model can be represented by two different types of functions: the probability density function (PDF) and the cumulative distribution function (CDF, which are expressed as below [10]:

$$f(k, c, u) = \frac{k}{c} (u/c)^k e^{-(u/c)^k} \tag{1}$$

$$F(k, c, u) = 1 - e^{-(u/c)^k} \tag{2}$$

where $f(u)$ is the probability of the observed wind speed $u$ (m/s), $c$ is the scale parameter with the same unit of wind speed, and $k$ is the shape parameter with the dimensionless unit. $F(u)$ represents the probability of all wind speeds less than $u$.

To calculate the average wind speed $\bar{u}$ and variance $\sigma$ of the known wind speed data, the following expressions can be used:

$$\bar{u} = \frac{1}{n} \left[ \sum_{i=1}^{n} u_i \right] \tag{3}$$

$$\sigma = \left[ \frac{1}{n\text{-}1} \sum_{i\text{-}1}^{n} (u_i - \bar{u})^2 \right]^{1/2} \tag{4}$$

Using Equations (3) and (4), Weibull parameters $k$ and $c$ can be calculated by the following equations:

$$k = (\sigma_{\overline{v}})^{-1.086} \ (1 \leq k \leq 10) \tag{5}$$

$$c = \frac{\bar{u}}{\Gamma \left( 1 - \frac{1}{k} \right)} \tag{6}$$

Where Γ is the gamma function, and using the Stirling approximation the gamma function of ($x$) can be expressed as follows:

$$\Gamma(x) = \int_0^\infty e^{-v} v^{x-1} dv \tag{7}$$

The Weibull shape and scale parameters provide critical inputs for turbine specification trade-offs. Shape parameter drives structural and control system design to handle wind variability, and scale parameter informs energy yield potential and rotor/generator sizing. A site-specific analysis integrating these parameters ensures optimal balance between energy production, capital costs, and operational reliability.

### 3.2 Most probable wind speed

For a given wind probability distribution, the most frequent wind speed is the most probable wind speed. Using the scale and shape parameters of the Weibull distribution function, the most probable wind speed can be calculated from the equation below.

$$V_{\text{mp}} = c \left( 1 - \frac{1}{k} \right)^{\frac{1}{k}} \ (m/s) \tag{8}$$

## 3.3 Air density calculation

During feasibility surveys for wind development, air density should be examined since wind energy is directly related to it. In this study, air density was determined using Equation (9) [29] by placing a barometer and a temperature sensor at 20 meters to measure air density.

$$\rho_{10min} = \frac{P_{10min}}{R_0 T_{10min}}$$

(9)

where $\rho_{10min}$ is the derived 10 min averaged air density, $T_{10min}$ is the measured absolute air temperature (K) averaged over 10 min, $P_{10min}$ is the measured air pressure (Pa) averaged over 10 min, and $R_0$ is the gas constant for dry air $(R_0 = 287.05$ J/kg K).

## 3.4 Wind power density

The power of the wind flowing through a blade sweep area ($A$) at a speed ($u$) can be expressed by the following equations:

$$P(v) = \frac{1}{2}\rho u^3 A$$

(10)

where $\rho$ is the air density The equation shows that wind power density increases proportionally to the cube of wind speed. Moreover, the calculation of wind power density (WPD) using in situ wind speed measurements can be performed with the following equation, which involves Weibull distribution analysis [30].

$$\frac{P}{A} = \int_0^\infty \frac{1}{2}\rho u^3 f(u)du = \frac{1}{2}\rho c^3 \Gamma\left(\frac{k+3}{k}\right)$$

(11)

By knowing the wind power density shown in Equation (11), wind energy density can be estimated by the following equation for the desired time, $T$,

$$\frac{E}{A} = \frac{1}{2}\rho c^3 \Gamma\left(\frac{k+3}{k}\right) T$$

(12)

When the wind speed frequency distribution is different, Equation (12) can be used to calculate the available wind energy for any specific period.

## 3.5 Wind shears

Since wind speed varies with height, anemometers at different observation stations are placed at different elevations. Therefore, before conducting any analysis, it is crucial to standardize the recorded wind speed data to a uniform height. The wind power law is widely regarded as an effective method, commonly used for evaluating wind energy, which requires adjusting wind speed data from differing heights to a consistent height.

To determine the wind shear exponent α when utilizing wind speeds measured at over two heights, apply the following power-law:

$$u_i = \beta(z_i)^\alpha$$

(13)

where, $u_i$ is the average wind speed at a height, $z_i$, $\beta$ is a constant.

By taking the natural logarithm of both sides of Eq. (13), The following equation is obtained,

$$\ln(u_i) = \alpha \ln(z_i) + \ln\beta$$

(14)

Applying the linear least squares regression to obtain the best fit line, thereby obtaining the optimal fitting value of $a$.

### 3.7 Turbulence intensity

Because of the cyclic loading on wind turbines, fatigue caused by aerodynamic forces can severely affect their structural integrity. When designing the turbines, it is essential to consider the Turbulence Intensity (TI) in relation to fatigue loads is crucial. Turbulence intensity (TI) is defined as the ratio of the standard deviation to the mean wind speed $\overline{u}$,

$$TI = \sigma / \overline{u} \tag{15}$$

### 3.8 Measure–Correlate–Predict method

Besides turbulence intensity, the IEC recommends creating wind turbine classes that account for EWS [29,31]. Extreme wind speeds are categorized into three types—Class I, II, and III in IEC 61400−1. each manufacturer's turbine has a different hub height, the design must accommodate the appropriate aerodynamic load by calculating the extreme wind speed with 1-year recurrence period and with 50-year recurrence period. However, in practical cases, the long-term wind speed observation records are lack. To overcome these drawbacks, one or more reference sites have been chosen and the relationship between the target and reference sites have been constructed using statistical methods MCP methods. MCP is widely used in renewable energy to extrapolate short-term site-specific measurements using long-term reference data. Some effective MCP methods have been proposed that used different types of functions to model correlation, such as linear regression model [32], support vector machine model [33], ANN [34], probabilistic method [35]. Traditional MCP methods rely on linear regression or parametric models, but these often fail to capture complex nonlinear relationships between variables. GAM address this limitation by integrating flexible smoothing functions, making them ideal for MCP analysis in spatially and temporally heterogeneous environments. GAM decompose the relationship between variables into additive smooth functions:

$$g(\mathrm{E}[Y]) = \beta_0 + f_1(x_1) + f_2(x_2) + \cdots + f_p(x_p) \tag{16}$$

where $g$ is a link function (e.g., Gaussian or Gamma), and $f_i$ are smooth terms modeled via splines or local polynomials. $Y$ represents the target site's wind speed, while includes reference site data, temporal variables (e.g., seasonality), and spatial covariates. Traditional MCP models poorly represent nonlinear interactions between mesoscale weather systems and local topography, leading to ±20% errors in extreme wind quantile estimation. GAM outperform linear models in capturing wind shear nonlinearities and turbulence-vegetation interactions, reducing prediction errors by 15–30% in coastal regions [36]. In this study, the GAM model is used for predicting 30-year wind speed time series, and then wind extreme analysis is done.

## 4. Results and discussion

### 4.1 Statistical analysis of wind variation

Wind statistics analysis, including turbulence intensity and wind shear, provide insights into the frequency of various wind speeds at a specific site with a specified average wind velocity. This data helps in selecting a wind turbine with the optimal cut-in speed (the wind speed at which the turbine starts generating usable energy) and cut-out speed (the wind velocity at which the turbine hits the maximum generator capacity, beyond which an increase in wind speed will not result in additional power output).

 **a) Diurnal wind variations.** The understanding power capacities in small, isolated power systems requires both the monthly or seasonal probability distribution of wind speeds and an evaluation of at least hourly probabilities of these

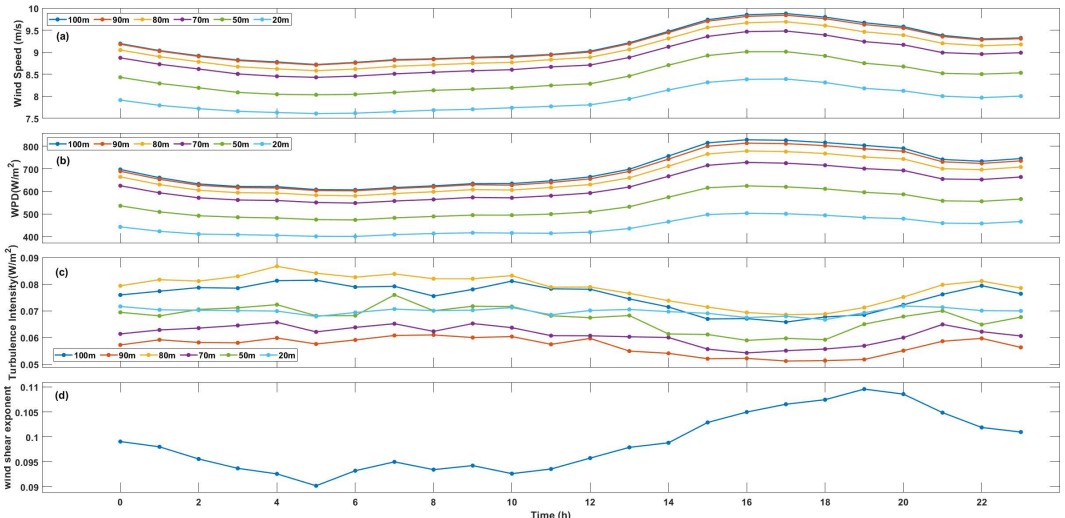

**Fig 2. Diurnal variation of wind speed. (a)**, wind power density (b) and turbulence intensity (c) at different height, and diurnal variation of wind shear exponent (d) at XiaPu station.

wind speeds. Diurnal variation of wind speed, wind power density and turbulence intensity at different height, and diurnal variation of wind shear exponent at XiaPu station is shown in Fig 2. Hourly wind speeds indicate that near-shore wind speeds in Fujian typically exhibit a bimodal variation. During morning surge (02:00–12:00), wind speeds increase by 15–20% compared to nighttime averages. During afternoon peak (14:00–20:00), maximum wind speeds occur, often exceeding 9 m/s above 70m height. Speeds gradually decrease after 00:00, with minimum values occurring before dawn. Wind speed differences between daytime peaks and nighttime lows average 2–3 m/s, reaching 4–5 m/s during winter monsoon periods. This diurnal signature is most pronounced in summer when thermal forcing dominates over synoptic systems. Such wind modal is related to Land-Sea thermal forcing. Solar heating intensifies land-sea temperature contrast during daytime, generating sea breezes that amplify wind speeds. The coastal terrain funnels these thermally-driven flows, creating acceleration zones. At night, reversed land breezes interact with background monsoon winds, causing partial speed reduction. Secondly, the Taiwan Strait's "wind tunnel" effect mechanically accelerates northwest monsoon winds. This effect peaks when solar heating strengthens pressure gradients across the strait. In addition, daytime convective mixing transports momentum from upper levels downward, enhancing surface winds. Stable nighttime atmospheric stratification inhibits this process.

Hourly WPD at XiaPu station is shown in Fig 2b, the variation of WPD is closely with wind speed, and it is reasonable that WPD depends on air density and cube of wind speed (Eq.10), while variation of air density is less than the wind speed. Fig 2c shows the turbulence intensity calculated according to Eq.15. The variation of TI shows a reversal pattern compared with wind speed, i.e., it is high TI values during low-speed period, and low TI value during high-speed period, such condition is benefit for wind power. Wind shear exponent (α) is given in Fig 2d, the variation of α is basically consistent with wind speed, but there is a little phase difference.

For PingTan station, the hourly wind speed, WPD, TI and WSE are shown in Fig 3. Their variation is similar with XiaPu station. The average hourly wind speed exceeds 5 m/s, and with an increase in measurement height, the wind speed keeps high throughout the entire day. At 100m, the daily maximum average wind speeds recorded were 9.88 m/s in XiaPu and 10.77 m/s in PingTan. During the observation period, both locations reached their peak average wind speed at 5 p.m. The significant fluctuations in load duration curves associated with electricity demand during the day indicate that energy

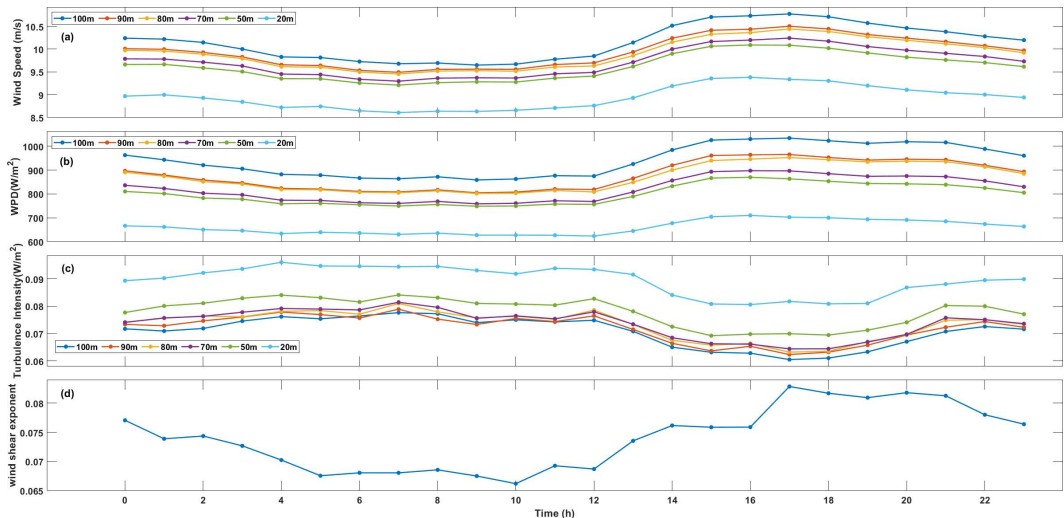

**Fig 3. The same as Fig 2, but for PingTan station.**

demand typically increase during daylight hours, showing a strong correlation between the demand curve and the local wind patterns.

**b) Monthly wind variations.** Significant variations in seasonal winds, especially the monsoon's impact on China's coastal areas, can be observed. The monthly fluctuations in wind speed are calculated through Eq. (3). Figs 4 and 5 give these monthly wind speed variations. Mean monthly wind speeds at two stations show notable similarities. Most monthly average wind speeds range between 8 m/s to 11 m/s, although some exceed 11 m/s and others are below 8 m/s. Overall, wind speeds tend to be higher in autumn and winter compared to other seasons, with spring showing the lowest speeds. XiaPu's peak average wind speed, about 11 m/s, occurs between October and November. In contrast, May records the

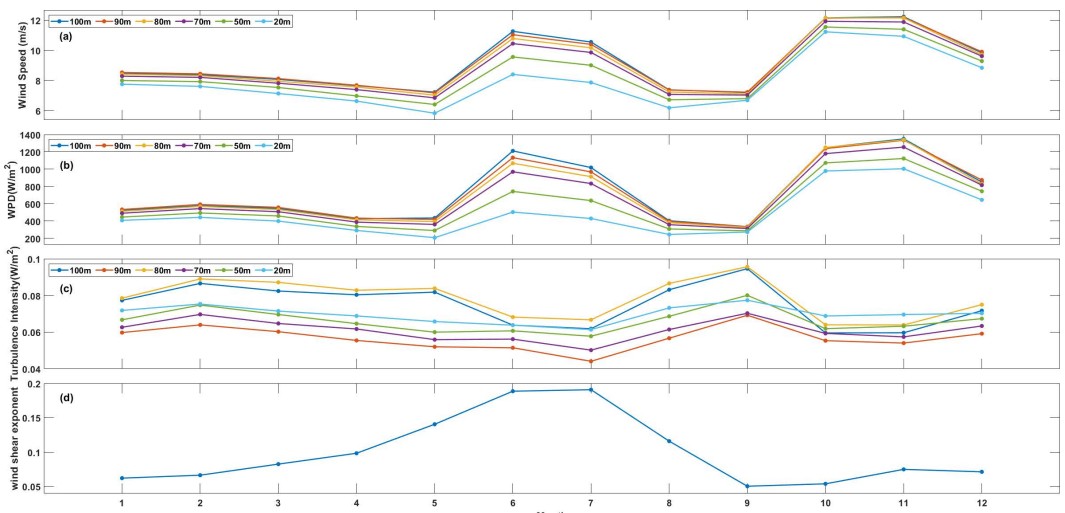

**Fig 4. Monthly variation of wind speed. (a),** wind power density (b) and turbulence intensity (c) at different height, and diurnal variation of wind shear exponent (d) at XiaPu station.

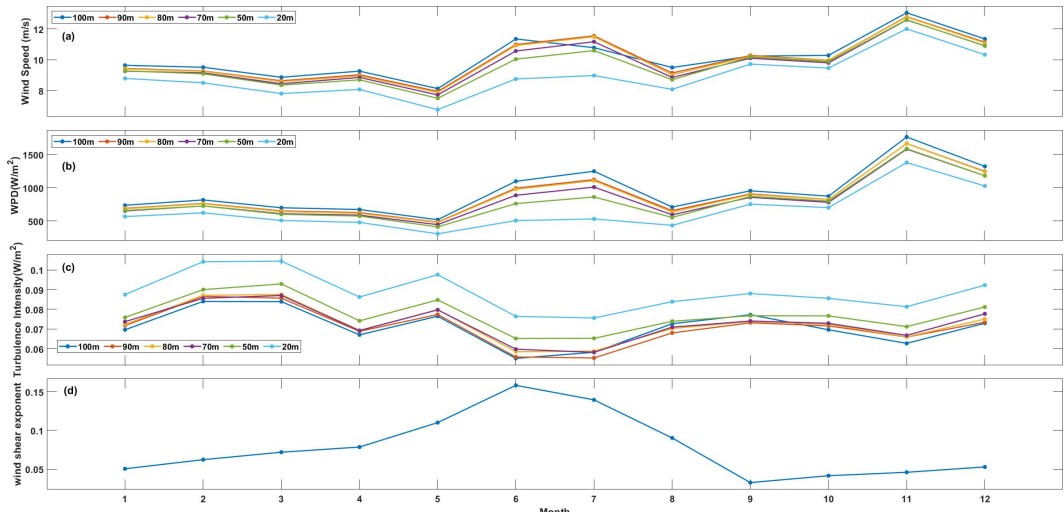

**Fig 5. The same as Fig 4, but for PingTan station.**

lowest average wind speed, around 8 m/s, at two monitoring stations. These variations may be due to strong Siberia in winter or the typhoon effects in summer. At XiaPu station, wind speed standard deviations at 100 m height range from 2.96 to 4.51 m/s, at 50 m from 2.81 to 3.83 m/s, and at 20 m from 2.67 to 3.65 m/s. This suggests that the area is well-suited for large-scale electricity generation using current wind turbine technology year-round.

The notable monthly variation highlights the necessity of differentiating between various months of the year when evaluating or designing a wind power project. Spring in Fujian exhibits pronounced wind speed fluctuations with larger TI due to seasonal transitions and regional meteorological dynamics. For instance, wind speeds can drop below 6 m/s during May, while episodic gusts from monsoons or typhoon precursors may exceed 15 m/s. Furter statistical analysis reveals that calm period (wind speed < 3 m/s) account for ~30% of spring days in Fujian, significantly reducing the effective power generation window. Wind turbines typically require a minimum cut-in speed of 3–4 m/s. During low-speed months, Fujian's CF drops to 15–20%, compared to 35–40% in high-wind seasons (summer and autumn). This seasonality necessitates overcapacity design or hybrid energy systems to stabilize supply, such as integrating solar or tidal energy to compensate for wind intermittency.

To ensure consistent energy production with lower wind speed period (e.g., in May), Some mitigation strategies should be implemented: 1) Technical optimization of wind turbines, deploy turbines with larger rotor diameters (e.g., 140–150m) and lower-rated capacities to capture low-speed wind energy more efficiently. For example, Goldwind's GW115/2.0 model increased annual output by 10.6% at 100m hub heights compared to 85 m towers in low-wind regions. Use hybrid steel-concrete or full-steel flexible towers (120–140m) to exploit higher wind speeds at elevated heights, consider the positive wind shear coefficients in the area. Optimize power tracking across low-to-medium wind speeds (2.5–5 m/s) through permanent magnet generators, achieving near-optimal power coefficient throughout the entire operational range. 2) Energy storage integration, deploy battery storage systems to store excess energy during high-wind periods and discharge during low-wind intervals, smoothing output fluctuations. Implement pumped hydro or hydrogen electrolysis systems for multi-day energy reserves. 3) Constructing multi-energy complementary systems, integrate photovoltaic panels to compensate for wind intermittency. For instance, solar generation typically peaks in summer, offsetting reduced wind output.

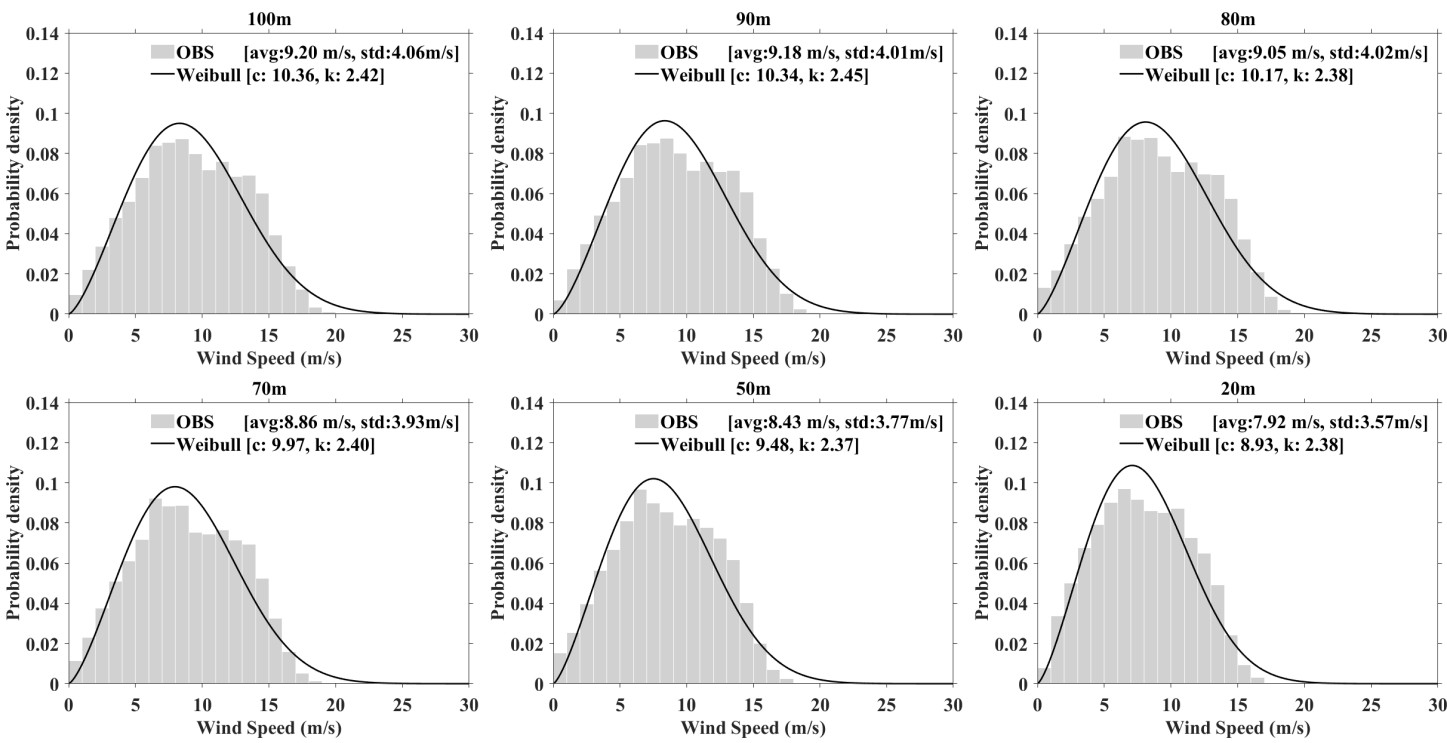

**Fig 6. Observed wind speed probability density distribution and fitted Weibull curve at XiaPu station.**

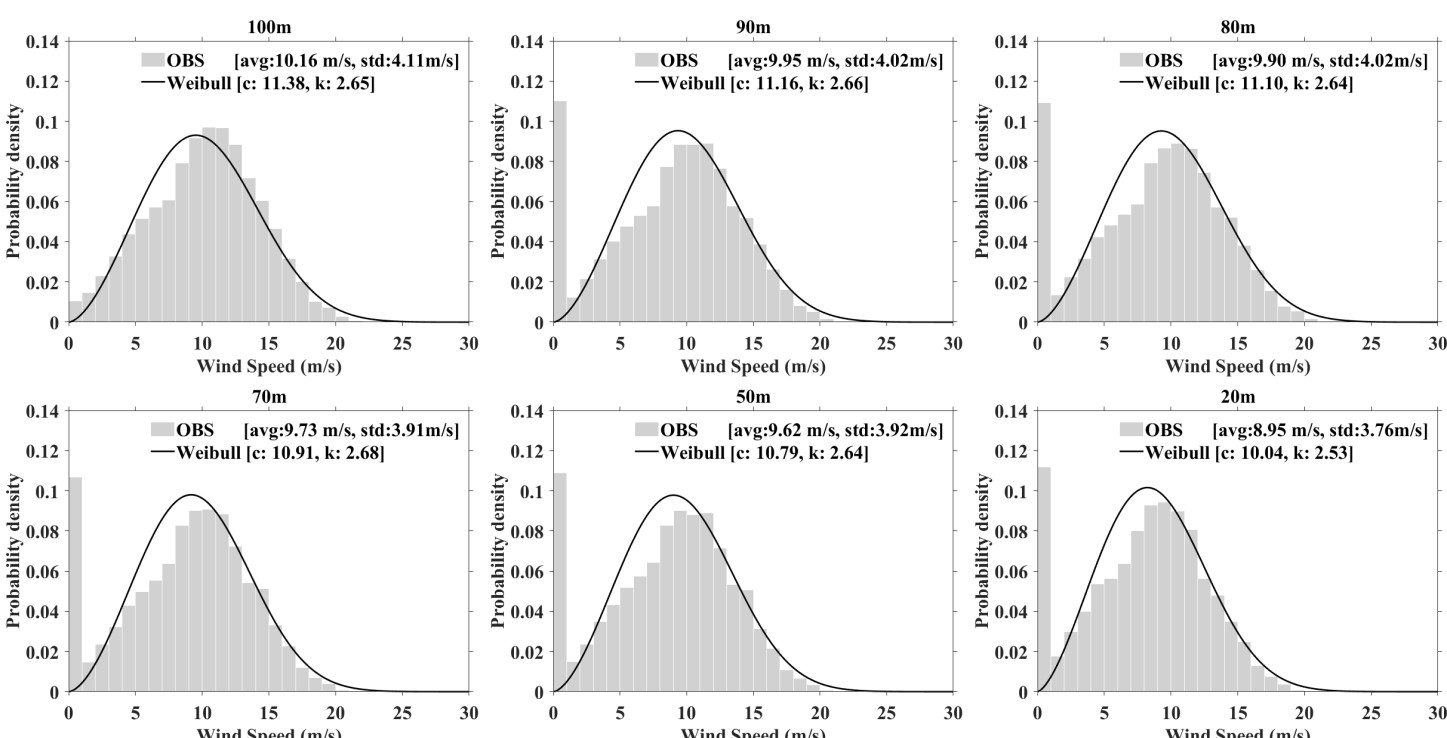

**Fig 7. The same as Fig 6, but for PingTan station.**

**c) Weibull distribution and variations of parameters.** The Weibull distributions for six different heights at both stations are shown in Figs 6 and 7. The histogram represents the frequency of actual wind speeds, while the Weibull distribution curve, estimated using the maximum likelihood method, demonstrates a strong correlation. The annual average wind speeds at various heights are 9.2 m/s, 9.18 m/s, 9.05 m/s, 8.86 m/s, 8.43 m/s, and 7.82 m/s, respectively. As the measurement height increases, higher wind speeds are observed. Concurrently, the scale parameter ranges from 8.93 m/s to 10.36 m/s and increases with height. The shape parameter remains around 2.4, with slight variations across different height.

Table 3 highlights the differences in the standard deviation, shape parameter, and scale parameter as determined by Equations (4)–(7). Using the 100m height at PingTan station as an example, the monthly standard deviation varies between 3.29 and 4.56 m/s. The shape parameter ranges from 2.28–3.50, while the scale parameters range from 9.16 to14.49 m/s. The mean values for the shape and scale parameters are 2.86 and 11.35 m/s, respectively.

The shape parameter (k) of the Weibull distribution reflects the variability and skewness of wind speed data, low k (<2) indicates high variability with frequent low-to-moderate wind speeds and a long right tail (extreme wind events). Turbines in such regions require robust structural designs to withstand intermittent high-stress conditions, prioritizing durability over peak efficiency. High k (>3) suggests stable wind regimes with concentrated wind speeds near the scale parameter. These favors turbines optimized for narrow operational ranges (e.g., higher cut-in speeds) to maximize energy capture within the dominant wind band.

The scale parameter (c) approximates the characteristic wind speed. When scale parameter increases with height, higher altitudes typically exhibit stronger and more consistent winds. Turbines at greater heights should prioritize larger rotor diameters and higher-rated power capacities to exploit elevated scale parameter. For low scale parameter (<6 m/s), requires turbines with low cut-in speeds (1.5–3 m/s) and variable-pitch blades to optimize performance in suboptimal wind regimes.

Generally speaking, the scale parameters increase with height. The scale parameter of the Weibull distribution reflects the characteristic wind speed magnitude. Its increase with height primarily stems two seasons. Firstly, Wind shear diminishes with altitude due to decreased ground roughness, allowing higher wind speeds. Secondly, the vertical wind profile typically follows a power-law relationship (Eq.13, Figs 11 and 12), causing scale parameter to scale proportionally with height. Although the general trend of parameter increasing with height holds globally, its magnitude varies regionally due to different terrain and atmospheric stability condition.

## 4.2 Statistical analysis of air density

Fig 8 shows the calculated air density histograms at 20m height. The Fig 6a represents air density histogram at XiaPu station, while the Fig 6b represents air density histogram at PingTan station. Air density show the highest probability distributed between 1.16 and 1.17 kg/m³. The average air density at XiaPu station is 1.21 kg/m³, and at PingTan station, it is 1.20 kg/m³. These values are 1.2% and 2.0% lower than the standard air density of 1.225 kg/m³, respectively. Compared to other areas, energy production at the demonstration wind farm is expected to decrease even when using the same wind turbines. Considering wind shear exponent and air density, increasing the hub height of wind turbines at the candidate site is advantageous as it can boost annual energy production, but it also will increase construction costs.

## 4.3 Wind power and energy density

Wind rose diagrams are indeed critical for analyzing the spatial distribution of wind energy potential across directional sectors. Wind roses segment wind data into 16 directional sectors, quantifying both occurrence frequency and average/peak wind speeds for each sector. For example, a sector with 30% frequency and 8 m/s average speed indicates higher energy yield potential compared to a sector with 10% frequency and 5 m/s speed. By integrating wind speed distributions (e.g.,

**Table 3. Monthly standard deviation and Weibull parameters (k, c).**

| Month | Parameters | XiaPu Station | | | | | | PingTan Station | | | | | |
|---|---|---|---|---|---|---|---|---|---|---|---|---|---|
| | | 100m | 90m | 80m | 70m | 50m | 20m | 100m | 90m | 80m | 70m | 50m | 20m |
| Jan | $\sigma$ | 3.04 | 3.08 | 3.05 | 3.01 | 2.96 | 2.88 | 3.29 | 3.23 | 3.22 | 3.14 | 3.18 | 3.08 |
| | c | 9.48 | 9.53 | 9.4 | 9.26 | 8.94 | 8.68 | 10.7 | 10.48 | 10.45 | 10.29 | 10.31 | 9.78 |
| | k | 3.08 | 3.05 | 3.02 | 3.02 | 2.95 | 2.96 | 3.26 | 3.27 | 3.26 | 3.3 | 3.27 | 3.17 |
| Feb | $\sigma$ | 3.69 | 3.73 | 3.71 | 3.63 | 3.53 | 3.43 | 4.11 | 4.06 | 4.04 | 3.93 | 4.01 | 3.99 |
| | c | 9.43 | 9.5 | 9.33 | 9.21 | 8.89 | 8.57 | 10.65 | 10.37 | 10.36 | 10.26 | 10.2 | 9.54 |
| | k | 2.43 | 2.42 | 2.35 | 2.38 | 2.35 | 2.38 | 2.44 | 2.39 | 2.42 | 2.49 | 2.4 | 2.21 |
| Mar | $\sigma$ | 3.78 | 3.79 | 3.81 | 3.73 | 3.63 | 3.52 | 4.08 | 4 | 4 | 3.9 | 3.95 | 3.84 |
| | c | 9.09 | 9.17 | 8.94 | 8.81 | 8.5 | 8.06 | 9.96 | 9.71 | 9.65 | 9.51 | 9.4 | 8.77 |
| | k | 2.24 | 2.28 | 2.18 | 2.2 | 2.19 | 2.15 | 2.28 | 2.25 | 2.25 | 2.28 | 2.21 | 2.09 |
| Apr | $\sigma$ | 3.26 | 3.28 | 3.24 | 3.17 | 3.14 | 2.97 | 3.36 | 3.32 | 3.31 | 3.2 | 3.3 | 3.25 |
| | c | 8.6 | 8.63 | 8.49 | 8.29 | 7.81 | 7.46 | 10.34 | 10.08 | 10.01 | 9.9 | 9.75 | 9.08 |
| | k | 2.49 | 2.47 | 2.45 | 2.44 | 2.28 | 2.36 | 3.02 | 2.97 | 2.95 | 3.05 | 2.89 | 2.7 |
| May | $\sigma$ | 3.71 | 3.62 | 3.58 | 3.45 | 3.17 | 2.75 | 3.64 | 3.53 | 3.55 | 3.44 | 3.38 | 3.1 |
| | c | 8.17 | 8.12 | 7.93 | 7.74 | 7.24 | 6.59 | 9.16 | 8.96 | 8.89 | 8.72 | 8.46 | 7.63 |
| | k | 2.07 | 2.11 | 2.07 | 2.1 | 2.13 | 2.26 | 2.36 | 2.39 | 2.35 | 2.39 | 2.34 | 2.29 |
| Jun | $\sigma$ | 4.51 | 4.38 | 4.36 | 4.21 | 3.84 | 3.35 | 3.64 | 3.5 | 3.5 | 3.39 | 3.22 | 2.8 |
| | c | 12.61 | 12.35 | 12.06 | 11.69 | 10.7 | 9.42 | 12.55 | 12.16 | 12.08 | 11.7 | 11.12 | 9.7 |
| | k | 2.74 | 2.78 | 2.68 | 2.71 | 2.73 | 2.77 | 3.51 | 3.54 | 3.52 | 3.53 | 3.52 | 3.5 |
| Jul | $\sigma$ | 4.35 | 4.25 | 4.21 | 4.09 | 3.74 | 3.31 | 3.81 | 3.52 | 3.52 | 3.36 | 3.17 | 2.75 |
| | c | 11.85 | 11.67 | 11.41 | 11.06 | 10.1 | 8.84 | 11.99 | 12.69 | 12.62 | 12.28 | 11.64 | 9.88 |
| | k | 2.66 | 2.69 | 2.63 | 2.63 | 2.61 | 2.59 | 3.16 | 3.74 | 3.72 | 3.83 | 3.84 | 3.65 |
| Aug | $\sigma$ | 3.51 | 3.44 | 3.43 | 3.37 | 3.23 | 3.01 | 4.12 | 3.63 | 3.6 | 3.49 | 3.34 | 3.02 |
| | c | 8.31 | 8.3 | 8.14 | 7.95 | 7.54 | 6.97 | 10.68 | 10.27 | 10.15 | 9.95 | 9.77 | 9.05 |
| | k | 2.19 | 2.25 | 2.19 | 2.17 | 2.13 | 2.16 | 2.42 | 2.69 | 2.67 | 2.71 | 2.79 | 2.86 |
| Sep | $\sigma$ | 3.02 | 2.96 | 3.04 | 2.96 | 2.93 | 2.86 | 3.97 | 3.84 | 3.84 | 3.74 | 3.72 | 3.55 |
| | c | 8.06 | 8.1 | 7.96 | 7.89 | 7.62 | 7.51 | 11.44 | 11.5 | 11.45 | 11.3 | 11.39 | 10.86 |
| | k | 2.51 | 2.62 | 2.46 | 2.53 | 2.45 | 2.5 | 2.76 | 2.89 | 2.87 | 2.92 | 2.97 | 2.98 |
| Oct | $\sigma$ | 2.96 | 2.91 | 2.98 | 2.89 | 2.81 | 2.67 | 3.8 | 3.68 | 3.69 | 3.59 | 3.63 | 3.47 |
| | c | 13.19 | 13.16 | 13.19 | 12.95 | 12.54 | 12.17 | 11.48 | 11.14 | 11.14 | 10.95 | 11.04 | 10.57 |
| | k | 5.06 | 5.16 | 5.02 | 5.11 | 5.09 | 5.3 | 2.98 | 2.99 | 2.98 | 3.02 | 3.01 | 3.02 |
| Nov | $\sigma$ | 3.37 | 3.36 | 3.47 | 3.41 | 3.4 | 3.38 | 4.36 | 4.28 | 4.31 | 4.19 | 4.22 | 4.06 |
| | c | 13.42 | 13.37 | 13.35 | 13.06 | 12.55 | 12.06 | 14.49 | 14.21 | 14.19 | 13.98 | 13.98 | 13.32 |
| | k | 4.4 | 4.38 | 4.2 | 4.17 | 3.97 | 3.8 | 3.48 | 3.47 | 3.43 | 3.49 | 3.46 | 3.41 |
| Dec | $\sigma$ | 3.83 | 3.89 | 3.85 | 3.88 | 3.83 | 3.65 | 4.56 | 4.46 | 4.5 | 4.42 | 4.45 | 4.34 |
| | c | 11.01 | 11.09 | 10.86 | 10.77 | 10.4 | 9.93 | 12.69 | 12.46 | 12.4 | 12.19 | 12.18 | 11.56 |
| | k | 2.81 | 2.77 | 2.72 | 2.68 | 2.59 | 2.63 | 2.68 | 2.7 | 2.64 | 2.63 | 2.62 | 2.53 |

Weibull parameters) with sector-specific data from wind roses, energy density (W/m²) can be estimated for each direction. This identifies dominant energy-contributing sectors.

Fig 9 shows the wind and energy roses at a height of 100 meters. These diagrams describe the distribution of directional frequency for both wind and WPD, calculated using equation (10). In Fig 9, the data is divided into eight directional sectors, revealing similar frequency patterns at both two observation stations. The predominant wind direction is from the northeast, while southwest winds occur less frequently.

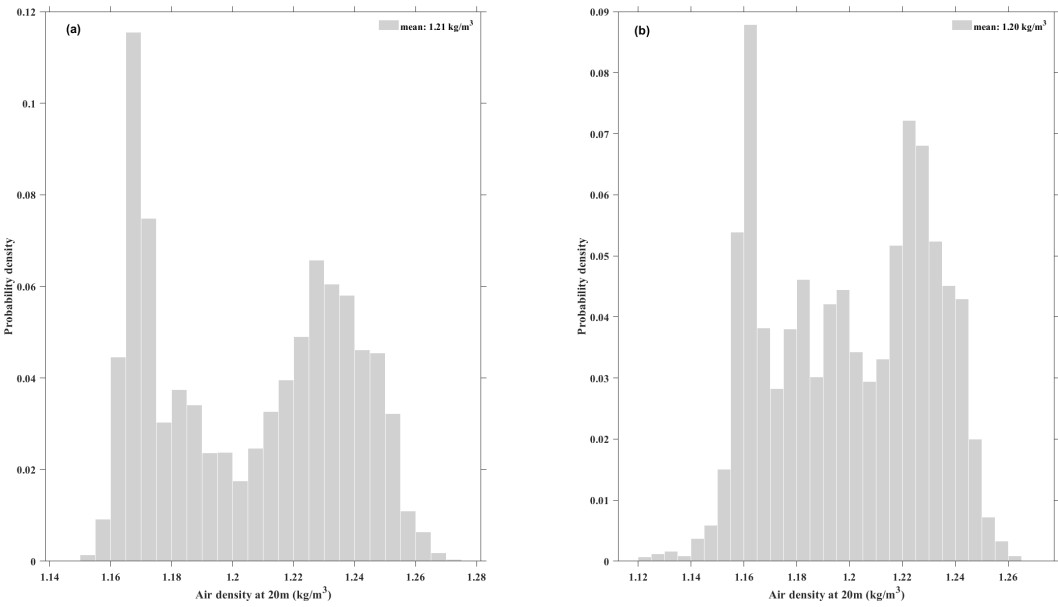

**Fig 8. Histogram of air density at. (a)** XiaPu station and **(b)** PingTan station.

**Fig 9. Wind speed.** (a) and wind power density (b) roses. First line indicates XiaPu station and second line indicates PingTan station.

**Table 4. Yearly mean wind speed, standard deviation and Weibull parameters (*k*, *c*), wind power density, wind energy density, most probable wind speed and wind speed carrying max energy.**

| Parameters | XiaPu Station | | | | | | PingTan Station | | | | | |
|---|---|---|---|---|---|---|---|---|---|---|---|---|
| | 100m | 90m | 80m | 70m | 50m | 20m | 100m | 90m | 80m | 70m | 50m | 20m |
| $\bar{u}$ | 9.2 | 9.18 | 9.05 | 8.86 | 8.43 | 7.92 | 10.16 | 9.95 | 9.9 | 9.73 | 9.62 | 8.95 |
| $u_{max}$ | 21.49 | 21.45 | 20.95 | 20.99 | 20.29 | 18.92 | 25.01 | 25.04 | 25.13 | 25.08 | 24.42 | 22.82 |
| $V_{mp}$ | 8.3 | 8.35 | 8.1 | 7.97 | 7.51 | 7.09 | 9.52 | 9.35 | 9.28 | 9.17 | 9.01 | 8.23 |
| $\sigma$ | 4.06 | 4.01 | 4.02 | 3.93 | 3.77 | 3.57 | 4.11 | 4.02 | 4.02 | 3.91 | 3.92 | 3.76 |
| $c$ | 10.36 | 10.34 | 10.17 | 9.97 | 9.48 | 8.93 | 11.38 | 11.16 | 11.1 | 10.91 | 10.79 | 10.04 |
| $k$ | 2.42 | 2.45 | 2.38 | 2.4 | 2.37 | 2.38 | 2.65 | 2.66 | 2.64 | 2.68 | 2.64 | 2.53 |
| P/A | 766.88 | 755.12 | 734.23 | 688.38 | 596.78 | 497.94 | 960.42 | 901.36 | 892.91 | 840.62 | 819.37 | 677.49 |
| E/A | 6717.9 | 6614.8 | 6431.8 | 6030.2 | 5227.8 | 4361.9 | 8413.3 | 7895.9 | 7821.9 | 7363.9 | 7177.7 | 5934.8 |

The wind field in China's seas experiences significant seasonal variations due to the monsoon climate, with higher wind energy levels in autumn and winter compared to spring and summer [37] (Li et al.; 2023). Table 4 details the annual average wind speed, standard deviation, Weibull parameters, the most probable wind speed, the wind speed with maximum energy, power density, and energy density. These values are derived from Equations (3)–(8) and (11)–(12). In 2020, XiaPu recorded a peak WPD of 1341.61 W/m², with a minimum average WPD of 766.88 W/m² at a height of 100m. The wind energy density values ranged from 3082.63 to 11753.52 kWh/m²/year. At a height of 100m in XiaPu, the most probable wind speed is 8.3 m/s, and the wind speed carrying maximum energy is 9.52 m/s. Due to the Betz limit, which states that no apparatus can convert all produced power into a usable form, The optimal power extraction coefficient for wind energy conversion systems is 0.593 [38]. Therefore, the maximum extractable power in XiaPu for 2020 is 208.05 W/m² per swept area of the wind turbine.

Wind energy density depends on the cube of wind speed, but vertical wind shear (variation in speed with height) and atmospheric stability alter the wind profile. For instance, stable marine boundary layers in Fujian's offshore regions may suppress turbulence, reducing energy extraction efficiency at turbine hub heights.

Fujian's coastal terrain (e.g., the Taiwan Strait funneling effect) amplifies wind speed intermittency. Seasonal monsoon shifts (e.g., winter northeasterlies vs. summer southwesterlies) create temporal mismatches between peak wind speeds and energy density due to turbulence and directional variability. Ocean-atmosphere interactions, such as sea surface temperature gradients and coastal upwelling, modify wind speed stability. For example, cold wakes from typhoons can temporarily reduce wind energy density despite sustained high wind speeds. ENSO (El Niño-Southern Oscillation) and Pacific Decadal Oscillation modulate Fujian's wind regimes. During La Niña, intensified northeasterlies elevate wind speeds but may increase shear-induced turbulence, reducing effective energy density.

### 4.4 Turbulence intensity

The cyclic loading of wind turbines due to, aerodynamic forces can significantly affect their structural integrity. turbulence intensity, characterized by rapid fluctuations in wind speed over short periods, is vital for assessing fatigue loads during turbine design. In classifying wind turbines are classified into three groups- A, B, and C, based on the intensity of environmental turbulence.

The impact of turbulence intensity variations across IEC classes (A/B/C) on wind turbine performance manifests through three key mechanisms, with amplified effects during typhoon seasons. Class A turbines (designed for TI ≤ 0.18 at 15 m/s) experience 40–60% higher fatigue loads compared to Class C (TI ≤ 0.12) under normal conditions, accelerating material fatigue in blades, gearboxes, and tower bases. During typhoons, TI often exceeds IEC class limits, particularly in Class C turbines optimized for low-TI environments. Typhoons generate extreme coherent gusts with simultaneous

TI surges. Class A turbines, while robust, still face 30–50% exceedance in tower base bending moments versus design loads, risking foundation failure. For typhoon-prone regions, hybrid Class A+/B designs with 10–15% increased safety margins on tower resonance frequencies are recommended.

TI is affected by factors such as average wind speed, surface roughness, atmospheric stability, and terrain features. It refers to the random fluctuations from the average wind speed at any given moment. Typically, the period lasts no more than one hour, but according wind energy engineering standards, it is often set to 10 minutes. By examining the time history of wind speed recorded at a sufficient resolution, key parameters can be identified. Fig 10 illustrates the ambient

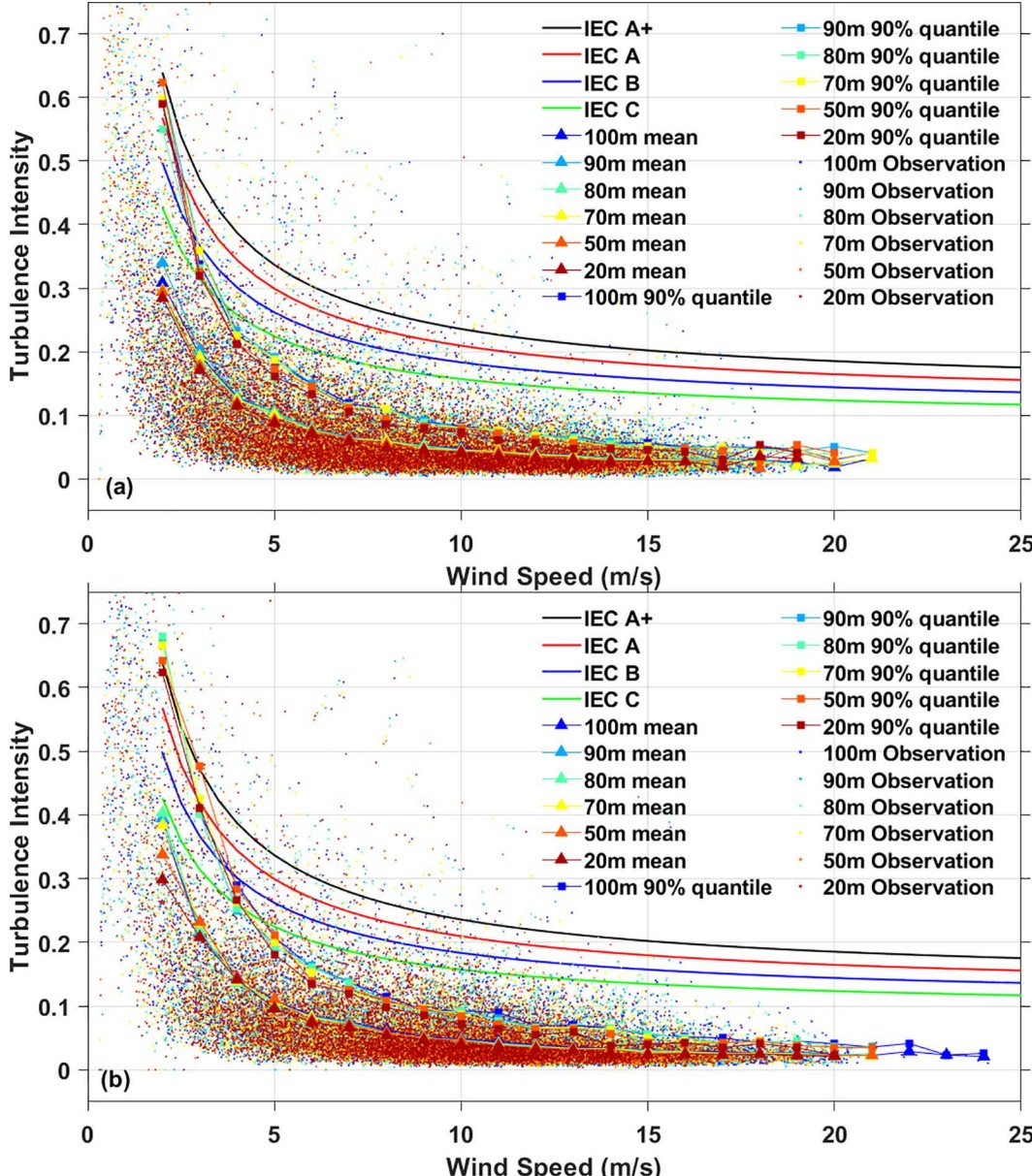

**Fig 10. Turbulence intensity analysis. (a)**XiaPu station, **(b)** PingTan station.

turbulence intensity as defined by the IEC standard compared to turbulence intensity from observations. In Fig 10, categorizes turbulence characteristics into Class A, Class B, and Class C, corresponding to extremely high, high, medium, and low according to IEC 61400−1. TI is measured using equation (15) and represented with dots in Fig 10, averaged over a bin interval of 1 m/s. Across all wind speeds, the average TI remained below 10%. The 90% quantile of TI also not meet the IEC Class A+ standard. Therefore, it is essential to choose wind turbines designed for high turbulence intensity, as the measured values tends to be relatively high. Beyond extreme wind speed, TI plays a crucial role in determining the most appropriate turbine.

Low turbulence intensity (typically defined as <10%) implies weaker wind speed fluctuations, which reduces dynamic fatigue loads on turbine components (e.g., blades, towers). The low turbulence intensity suggests the region is generally suitable for offshore wind farms due to lower operational risks. However, tailored turbine designs—particularly optimizing aerodynamics and wake management—are recommended to fully exploit the stable wind regime while addressing unique challenges like reduced shear and wake persistence. Reduced turbulence allows for lighter-weight materials in blades and towers, as fatigue resistance requirements relax. However, this necessitates trade-offs with durability under extreme weather events (e.g., typhoons) common in regions like the Fujian coastal area.

## 4.5  Wind shears

Fig 11 shows the variation in seasonal wind shears. The parameter $\alpha$ is derived from the wind speeds using Equation (14). During the one-year measurement period, $\alpha$ averaged 0.099 at XiaPu and 0.081 at PingTan, both lower than the value typical onshore value of 1/7, indicating conditions similar to those in open sea areas. Seasonal fluctuation showed that $\alpha$ was higher in spring and summer compared to other seasons, which had nearly identical values. During summer, the highest α value of 0.170, exceeds the previously reported offshore values of 0.11 or 0.12 in earlier studies [39–40]. Summer weather, marked by frequent rain and typhoons, increases surface roughness. Additionally, the α value is generally lower during day than at night due to more stable atmospheric condition at night and instability during the day.

The seasonal fluctuation of α in Fujian's offshore areas stems from multiple interacting factors:1) Fujian experiences strong winter northeasterly monsoons and summer southwesterlies, creating contrasting atmospheric stability. Winter monsoon-driven turbulence increases vertical mixing, reducing α (flatter wind profile), while summer stratification amplifies

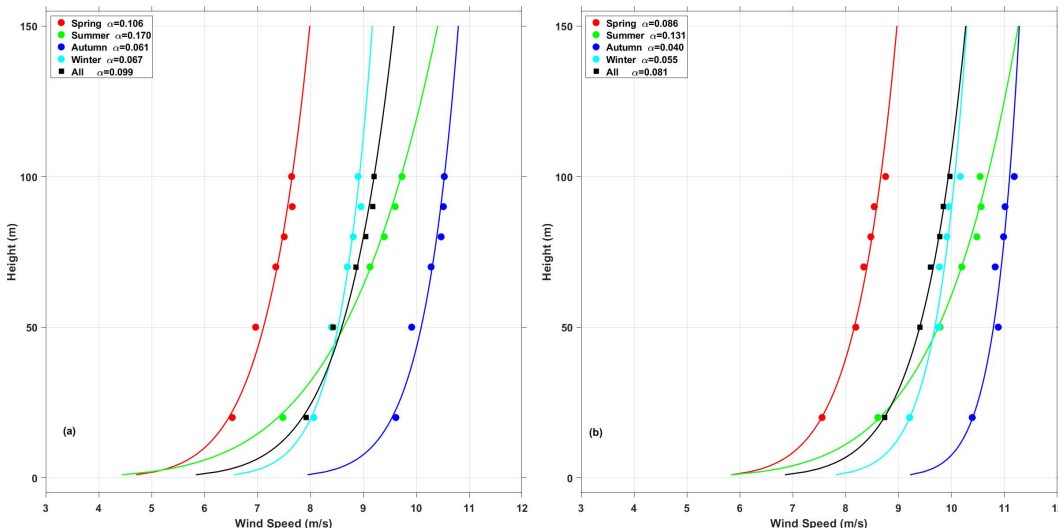

**Fig 11. The seasonal variation of the wind shear. (a)** XiaPu station, **(b)** PingTan station.

α. 2) Seasonal sea-land temperature gradients modify boundary layer stability. Summer solar heating intensifies thermal inversions, steepening wind speed gradients (higher α), whereas winter convective mixing flattens profiles. 3) Summer/ autumn typhoons induce extreme wind shear and transient α spikes, disrupting typical seasonal patterns. 4) Coastal upwelling in summer (e.g., Fujian's Min-Zhe Coastal Current) cools surface waters, enhancing near-surface wind acceleration and α variability.

Higher α in summer favors taller towers to capture stronger winds at elevated heights. Summer α > 0.25 exacerbates blade root bending moments by 18–25%, accelerating material fatigue. Lower α in winter increases wake recovery distances by 15–20%, necessitating 10–15% larger turbine spacing in predominant wind sectors. Rapid α shifts during typhoon transitions cause 5–10% power overestimation in standard IEC models, requiring real-time shear-adaptive control.

Fig 12 shows the variation in the wind shear exponent between average daytime and nighttime conditions. It is shows that the daytime wind shear exponent is slightly lower than nighttime. This difference is usually due to atmospheric instability caused by temperature variations with height. From this, it can be inferred that wind shear may cause higher fatigue loads during nighttime, as surface roughness tends to be less pronounced during the day compared to at night.

Lower daytime WSE indicates weaker vertical wind speed gradients. At 100m hub height, daytime wind speeds may increase only 1.2–1.5 m/s compared to 10m height, versus 2.0–2.8 m/s at night with higher WSE. This reduces daytime energy capture potential by 15–25% for standard turbines compared to nighttime operations. Reduced daytime WSE creates more uniform wind distribution across rotor planes, decreasing asymmetric blade loading.

## 4.6 Extreme wind speed

EWS during typhoons impose non-linear mechanical stresses on turbine components (e.g., blades, gearboxes, and towers). Typhoon-induced gusts exceeding 25 m/s can cause blade bending moments exceeding design limits, accelerating microcrack propagation. High turbulence intensity during typhoons increases gearbox vibration amplitudes by 30–50%, shortening lubrication cycles. These necessitates shorter inspection intervals (e.g., post-typhoon structural checks) and proactive replacement of fatigue-prone parts.

The IEC recommends categorizing wind turbine based on EWS and TI (IEC, Part1 and Part3). Manufacturers design wind turbines to endure aerodynamic loads corresponding to a reference extreme wind speed occurring every 50 years

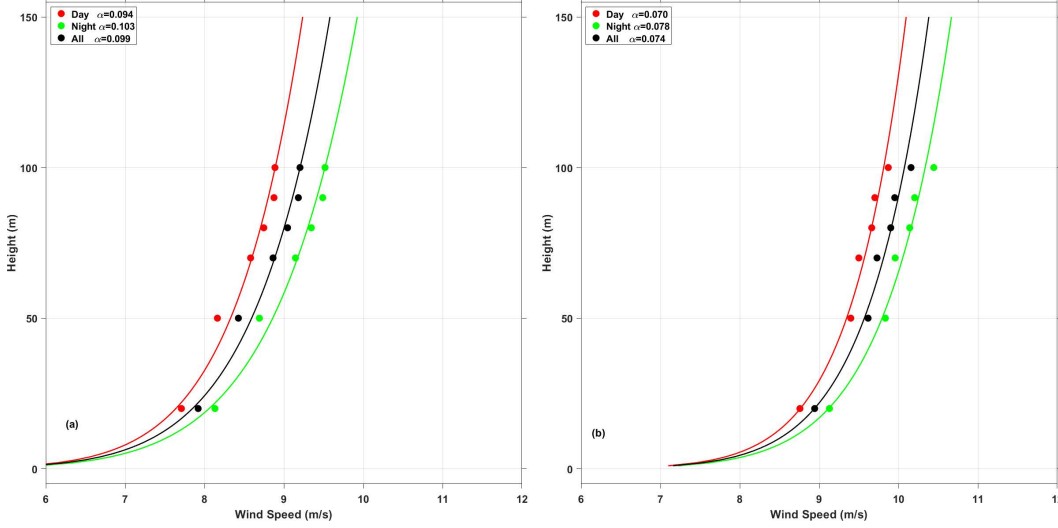

**Fig 12. The diurnal variation of the wind shear. (a)** XiaPu station, **(b)** PingTan station.

and a yearly recurrence at hub height. IEC 61400−1 defines a reference extreme wind speed, $V_{ref}$, for each class. $V_{ref}$ represents the extreme average wind speed over 10 minutes with a 50 years recurrence period at the height of the turbine hub. According to IEC 61400−1, extreme wind speeds are classified into three types: I, II, and III.

A wind measurement campaign using a tall meteorological mas for wind assessment typically lasts 1–3 years. However, the data collected during this period may not accurately reflect the wind conditions over 20-year lifespan of a wind turbine. Using long-term data for assessing extreme wind speeds can reduce estimation uncertainties. To forecast long-term wind conditions, the Measure-Correlate-Predict (MCP) technique, widely adopted in the wind power sector [41], combines several years of direct measurements with long-term reference wind data collected over more than a decade to forecast long-term wind conditions. In the present study, ERA5 data were collected over a 30-year period, from 1993 to 2023, at a height of 100 m, as depicted in Fig 1, and used as reference data.

Before applying the MCP method, it is essential to analyze the correlation between in situ data and concurrent reference data to ensure the reference data's validity. A linear regression analysis between the measurements and the corresponding ERA5 data is reveals a correlation coefficient of 0.69 (Figure omitted), considered acceptable for MCP implementation. GAM forcasts indicate significantly higher annual wind speeds from 2021 to 2023, compared to notably lower speeds recorded in 1997 and 2002. For XiaPu, the 30-year average wind speed reported by ERA5 is 8.49 m/s, contrasting with the predicted average of 8.94 m/s.

Wind speeds classified as extreme were determined using the Pearson Type III frequency distribution [42], as shown in Fig 13. The extreme wind speed for XiaPu, corresponding to a recurrence period of 50 years, is 38 m/s. Due to the frequent occurrence of typhoons, it is advisable to use higher-class turbines in the waters around FuJian.

Extreme winds accelerate structural fatigue in turbine blades and bearings. Maintenance intervals must shorten to address micro-cracks and delamination risks. Real-time sensor data (vibration, stress) combined with machine learning models can forecast component degradation under repeated typhoon loads. For example, turbines in typhoon-prone

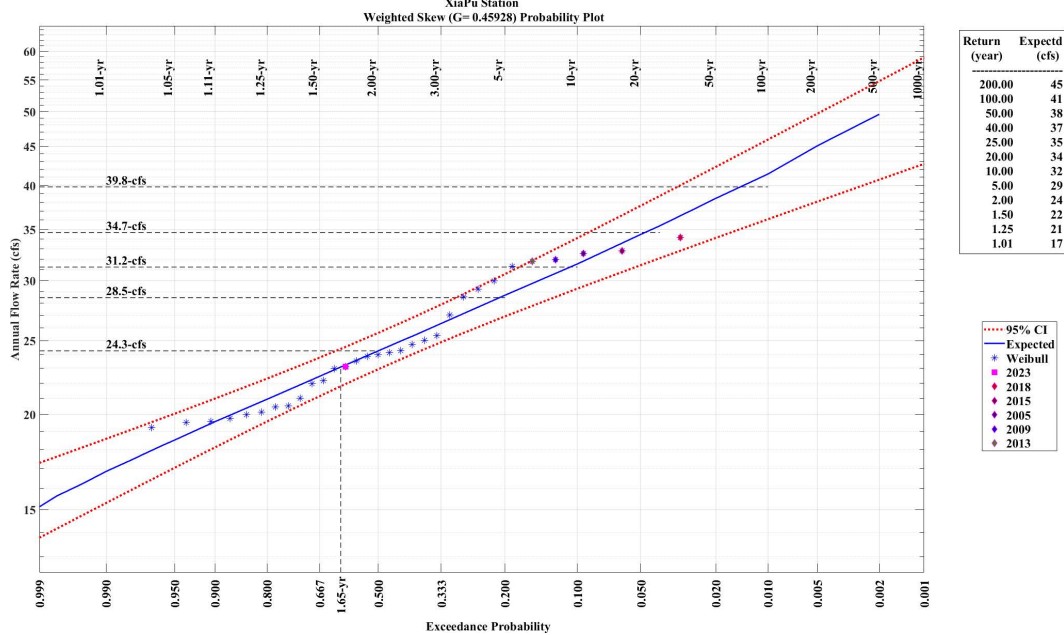

**Fig 13. Estimation of EWS with Pearson Type III frequency distribution fitted.**

regions may require 30% more frequent blade inspections than those in calmer areas. Sudden wind shear and turbulence during typhoons often damage pitch systems and yaw drives, necessitating immediate post-event inspections.

It should be noted that MCP methods rely on short-term correlations between reference and target sites to extrapolate long-term wind patterns. However, Extreme wind events are rare in short-term training datasets (e.g., < 5% of hourly data), causing sampling bias. For example, 50-year return period winds may have only 1–2 observed events in a 10-year reference period. In addition, extreme wind events (e.g., 99th percentile gusts) often exhibit non-stationary behavior over 30-year periods due to climate variability (e.g., ENSO cycles) and anthropogenic climate change. Studies show that MCP tends to underestimate extreme wind speeds by 8–15% in 30-year projections due to its linear regression assumptions and failure to capture tail-end distribution shifts.

## 5. Summary and conclusions

This study conducts a comprehensive assessment of offshore wind resource in the offshore region of Fujian, China based on 1-year wind profile record at two offshore meteorological masts. Combining the 30-year ERA5 reanalysis wind data, an adaptive MCP method is deployed to estimate the extreme wind speed. In the future, the design of demo-farms, turbines, and their foundations will be conducted using the estimated design basis of the abovementioned results. By quantifying Fujian's wind energy potential and its variability under climate change (e.g., typhoon frequency shifts), the study provides a template for other monsoon-dominated coastal regions in Southeast Asia and South America. This aligns with global efforts to standardize wind resource assessments for low-carbon transitions. The main findings are presented as follows:

The peak daily average wind speeds are prevailing between 12 a.m. and 11 p.m with lower turbulence intensity and higher wind shear exponent. The maximum daily average wind speeds recorded at a height of 100 m is large than 8.5m/s and 9.5m/s for XiaPu and PingTan, respectively. The maximum daily average wind speeds recorded at a height of 100 m were 9.88 m/s in XiaPu and 10.77 m/s in PingTan. Analysis of monthly average wind speeds revealed two distinct classifications. November typically experiences higher wind speeds compared to other months, and the second peak in wind power density occurs from May to August. The highest monthly average wind speeds at 100 m reached 12 m/s in both stations. Furthermore, the annual average wind speeds at 100 m, 50 m, and 20 m above mean sea level (AMSL) are recorded as 9.20 m/s, 8.43 m/s, and 7.92 m/s at XiaPu, respectively, whereas PingTan exhibits slightly lower average wind speeds.

Th wind rose indicates that the predominant directions for wind and energy recorded are northeast and southwest, reflecting typical monsoon climate characteristics. At a height of 100 m at XiaPu station, the expected wind speed is 8.3 m/s, wheäras the wind speed that maximum energy transmission is 9.52 m/s. In the year 2020, XiaPu recorded a maximum wind power density of 1341.61 W/m² at the height of 100 meters, with the minimum wind power density 351.90 W/m², while the average wind power density is 766.88 W/m². Additionally, the yearly wind energy density ranged from 3082.63 to 11753.52 kWh/m²/year.

The wind shear exponent and wind speed vary with the seasons influenced by monsoonal and the frequent summer typhoons. The 90th percentile of turbulence intensity was found to be below IEC Class A+ standard. Variable wind shear creates uneven blade loading, accelerating material fatigue in turbine components. For example, a 20% increase in wind shear variability can amplify blade root bending moments by 15–25%, reducing operational lifespan. Turbulence intensity fluctuations further exacerbate this through chaotic vortices, increasing vibration amplitudes by 30–50% in tower foundations.

Seasonality variation of wind shear exponent in the coastal of Fujian—rooted in monsoons, thermal gradients, and typhoons—demands adaptive turbine engineering and placement. Addressing these dynamics through height optimization, predictive controls, and typhoon-resilient designs is critical for maximizing offshore wind energy in this high-shear marine environment. Typhoons induce cyclic loading beyond design thresholds, reducing expected lifetimes by 5–10 years compared to standard projections. Regions with frequent typhoons now mandate turbines rated for IEC Class S (survival winds > 57 m/s), altering fatigue life calculations.

Based on the adaptive GAM, the 30-year wind speed is predicted. The 30-year predict average wind speed at a height of 100 meters is 10.84 m/s, measured at 100.0 m AMSL. The maximum wind speed expected over a 50-year return period is 38.0 m/s at that height, suggesting a wind turbine Class II rating. However, considering the surrounding turbulence intensity, it might be advisable to upgrade to IEC Class A+.

Based on the extreme wind speeds observed in Xiapu, the following adjustments to turbine specifications are recommended for typhoon-prone coastal regions high-risk zones (e.g., Fujian, Guangdong, Hainan) in China:1) Enhanced structural design standards. Upgrade turbine towers and blades to withstand wind speeds exceeding 70 m/s (Category 5 typhoon thresholds). For instance, Mingyang Smart Energy's MySE18.X-20MW turbines demonstrated resilience to extreme gusts through reinforced tower connections and adaptive blade pitching. Optimize monopile or jacket foundations with deeper embedment depths (e.g., 40–50 m for seabed penetration) to resist overturning moments caused by typhoon-induced lateral forces. 2) Using adaptive Control Technologies. Integrate AI-powered wind tracking systems to align turbines with rapidly shifting wind directions during typhoons, reducing lateral loads by 30–50%. Deploy hydraulic or electromechanical pitch systems to automatically feather blades when wind speeds exceed 25 m/s, preventing overspeed damage. In addition, extreme wind events like typhoons necessitate shorter maintenance cycles, advanced predictive algorithms, and revised lifetime models that account for cumulative fatigue and material limits. The integration of high-resolution wind data and adaptive engineering solutions is critical for turbine survivability in storm-prone regions.

It is important to note that this current work represents a preliminary study aimed at estimating the wind energy potential offshore in Fujian, China. The goal is to establish a comprehensive wind database and achieve accurate predictions before constructing and installing wind energy conversion systems. When evaluating wind power potential or selecting appropriate wind turbines, both wind data and site conditions such as terrain and varying heights must be taken into account. This consideration is crucial for the application of new wind energy generation technologies. In addition, extreme wind speed data fundamentally reshapes turbine maintenance and lifetime frameworks by highlighting typhoon-specific failure modes and enabling data-driven adaptation. Integrating multi-source monitoring with AI/ML models is critical to address typhoon recurrence challenges in Asia-Pacific typhoon-prone regions.

## Nomenclature

| | |
|---|---|
| MCP | Measure–Correlate–Predict |
| AMSL | Above Mean Sea Level |
| GAM | Generalized Additive Model |
| WPD | wind power density |
| WSE | Wind Shear Exponent |
| TI | Turbulence Intensity |
| GW | Giga Watt |
| MW | Mega Watt |
| IEC | International Electrotechnical Commission |
| ECMWF | European Centre for Medium-Range Weather Forecasts |
| ERA5 | ECMWF ReAnalysis 5th Generation |
| WRF | Weather Research and Forecasting Model |
| WWIII | Wave Watch III |
| MM5 | Mesoscale Model5 |
| GIS | Geographical Information Systems |
| CCMP | Cross-Calibrated, Multi-Platform |
| NCEP-CFSR | National Centers for Environmental Prediction-Climate Forecast System Reanalysis |
| PDF | Probability Density Function |

| CDF | Cumulative Distribution Function |
|-----|----------------------------------|
| EWS | Extreme Wind Speeds |
| ANN | Artificial Neural Networks |
| CF | Capacity Factor |

## Acknowledgments

This work has been supported by the National Key Research and Development Program of China (2022YFB4201400), the 111 Project of China (B18062). We are grateful to the reanalysis production center ECMWF and the Copernicus program for facilitating access to the reanalysis data.

## Author contributions

**Conceptualization:** Guanghong Liao.

**Data curation:** Wenfei Xue, Xiaokai Hu.

**Formal analysis:** Weidong Ji.

**Funding acquisition:** Rongfu Li, Weidong Ji, Qiaozhen Ning.

**Methodology:** Wenfei Xue, Xiaokai Hu.

**Project administration:** Rongfu Li.

**Resources:** Endi Zhai, Qiaozhen Ning.

**Software:** Endi Zhai.

**Supervision:** Guanghong Liao.

**Validation:** Hongying Yang.

**Visualization:** Hongying Yang, Xiaokai Hu.

**Writing – original draft:** Wenfei Xue.

**Writing – review & editing:** Guanghong Liao.

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
