## [Decision Letter · Decision Letter 0]

PONE-D-24-45330

Wind resource assessment Offshore Fujian using 30-year wind estimates

PLOS ONE

Dear Dr. Liao,

Thank you for submitting your manuscript to PLOS ONE. After careful consideration, we feel that it has merit but does not fully meet PLOS ONE’s publication criteria as it currently stands. Therefore, we invite you to submit a revised version of the manuscript that addresses the points raised during the review process.

We look forward to receiving your revised manuscript.

Kind regards,

Soheil Mohtaram

Academic Editor

PLOS ONE

Journal Requirements:

4. Please note that funding information should not appear in any section or other areas of your manuscript. We will only publish funding information present in the Funding Statement section of the online submission form. Please remove any funding-related text from the manuscript.

“National Key Research and Development Program of China (2022YFB4201400), the 111 Project of China (B18062)”

6. We note that your Data Availability Statement is currently as follows: “All relevant data are within the manuscript and in Supporting Information files.”

7. When completing the data availability statement of the submission form, you indicated that you will make your data available on acceptance. We strongly recommend all authors decide on a data sharing plan before acceptance, as the process can be lengthy and hold up publication timelines. Please note that, though access restrictions are acceptable now, your entire data will need to be made freely accessible if your manuscript is accepted for publication. This policy applies to all data except where public deposition would breach compliance with the protocol approved by your research ethics board. If you are unable to adhere to our open data policy, please kindly revise your statement to explain your reasoning and we will seek the editor's input on an exemption. Please be assured that, once you have provided your new statement, the assessment of your exemption will not hold up the peer review process.

8. We note that Figure 1 in your submission contain map/satellite images which may be copyrighted. All PLOS content is published under the Creative Commons Attribution License (CC BY 4.0), which means that the manuscript, images, and Supporting Information files will be freely available online, and any third party is permitted to access, download, copy, distribute, and use these materials in any way, even commercially, with proper attribution. For these reasons, we cannot publish previously copyrighted maps or satellite images created using proprietary data, such as Google software (Google Maps, Street View, and Earth). For more information, see our copyright guidelines: http://journals.plos.org/plosone/s/licenses-and-copyright.

9. We note you have included a table to which you do not refer in the text of your manuscript. Please ensure that you refer to Table 2 in your text; if accepted, production will need this reference to link the reader to the Table.

Additional Editor Comments:

Some sentences are quite long and complex, which can make them difficult to follow. Breaking them up into shorter, more concise sentences would improve readability.

The significance of choosing Fujian, China, as the study site isn't clear. Providing more context on why this location was selected would help readers understand its relevance.

The exact locations of Xiapu and Pintan should be specified. Including geographical coordinates or a brief description of these areas would provide better context.

Ensure consistency in the use of units throughout the paper. For example, wind speeds are given in m/s, and energy density in kWh/m²/year. Consistent use of units will make the data easier to compare and understand.

The methodology section could benefit from more detailed explanations. For instance, describe the types of wind measurement devices used and why they were chosen.

The introduction could be more engaging to capture the reader's interest. Consider starting with a compelling fact or statistic about the global energy crisis to draw readers in.

Explain why measurements were taken at specific heights (20, 50, 70, 80, 90, and 100 meters). This will help readers understand the rationale behind the data collection process.

The term "AMSL" (above mean sea level) should be defined when first used to ensure all readers understand its meaning.

Provide more context on the Measure-Correlate-Predict methodology. Explain how it works and why it was chosen for this study.

The transition between sections could be smoother. Adding transitional sentences or phrases would help guide readers through the paper more seamlessly.

Clarify the source and reliability of the ERA5 data. Discuss its accuracy and any potential limitations to give readers confidence in the data used.

The analysis of wind characteristics could be more detailed. For example, discuss the implications of average wind speed, wind direction frequency, and turbulence intensity on wind energy production.

Explain the significance of the 30-year span in the context of the study. Why was this period chosen, and how does it enhance the reliability of the findings?

The suitability of different wind turbine classes needs more justification. Provide data or references to support the selection of specific turbine classes for the offshore site.

Discuss the implications of extreme wind speeds in more detail. How do these speeds affect the design and operation of wind turbines?

The recommendation for wind turbine class II should be supported with more data. Explain why this class is suitable based on the study's findings.

Explain the criteria for upgrading to IEC Class A+. What factors were considered, and how do they impact the decision?

The conclusion could summarize the key findings more effectively. Highlight the most important results and their implications for offshore wind energy development.

Address any potential limitations of the study. Discuss how these limitations might affect the results and suggest areas for future research.

Consider the environmental impact of deploying wind turbines. Discuss any potential ecological effects and how they might be mitigated.

The language could be more concise and precise. Avoid unnecessary jargon and ensure that technical terms are clearly defined.

Ensure all technical terms are clearly defined. This will help readers who may not be familiar with the specific terminology used in the study.

The figures and tables should be referenced in the text. This will help readers understand how the visual data supports the study's findings.

The abstract should provide a clear overview of the study. Summarize the main objectives, methods, results, and conclusions in a concise manner.

Proofread for any grammatical or typographical errors. Ensuring the paper is free of errors will enhance its professionalism and readability.

Overall, the paper needs major revisions to improve clarity, coherence, and depth of analysis. Addressing these comments will strengthen the study and make it more accessible to a wider audience.

Reviewers' comments:

Reviewer's Responses to Questions

**Comments to the Author**

1. Is the manuscript technically sound, and do the data support the conclusions?

Reviewer #1: Yes

Reviewer #2: Partly

Reviewer #3: Partly

Reviewer #4: Yes

Reviewer #5: Yes

2. Has the statistical analysis been performed appropriately and rigorously? 

Reviewer #1: Yes

Reviewer #2: Yes

Reviewer #3: N/A

Reviewer #4: Yes

Reviewer #5: Yes

3. Have the authors made all data underlying the findings in their manuscript fully available?

Reviewer #1: Yes

Reviewer #2: Yes

Reviewer #3: Yes

Reviewer #4: Yes

Reviewer #5: Yes

4. Is the manuscript presented in an intelligible fashion and written in standard English?

Reviewer #1: No

Reviewer #2: Yes

Reviewer #3: No

Reviewer #4: No

Reviewer #5: No

5. Review Comments to the Author

Reviewer #1: Below are my comments regarding the manuscript:

- Abstract: The abstract requires grammatical revisions and careful proofreading for spelling errors. A clearer articulation of the main findings would enhance its effectiveness.

- Introduction: The introduction section needs a comprehensive revision. Currently, it lacks focus and should be condensed to present the key points more succinctly.

- References: The manuscript would benefit from the inclusion of more recent references. For instance, the following article discusses the statistical properties (mean, variance, and standard deviation) of wind turbine blades and could provide valuable insights: https://doi.org/10.1177/08927057231154427.

- Conclusion: The conclusion should be enhanced to provide a clearer overview of the scope of the work and the key achievements. A narrative format rather than a bullet-point list would improve clarity and coherence.

- Formatting of Equations: Consistency in the font used for all mathematical formulations is necessary. Please ensure that all formulas are uniformly formatted and revised.

- Overall Framework: The overall structure of the article requires significant revision. A more logical flow of information would improve readability and comprehension.

- Writing Quality: Please pay attention to the overall writing quality and grammar throughout the manuscript. Enhancing these aspects will strengthen the presentation of your arguments.

- Figure and Table Captions: All captions for figures and tables should be carefully revised for clarity and completeness. Ensure they accurately reflect the content.

- Figure 10: The explanation accompanying Figure 10 should be expanded. Currently, it is not sufficiently clear, and a more detailed description would help convey the intended message effectively.

Reviewer #2: Comments for authors:

Key Areas Needing Revision:

Main Gaps:

1. Seasonal and Temporal Variability in Wind Energy Density: While the paper provides a thorough analysis of wind speed and its relationship with energy density, it does not fully address how the temporal and seasonal fluctuations in wind energy density might impact the long-term feasibility and efficiency of offshore wind farms. The variations in seasonal wind energy density suggest that energy production may fluctuate more significantly than wind speed alone indicates, yet the study lacks a detailed analysis of these fluctuations in the context of wind farm operations over multiple years.

2. Effect of Typhoons on Wind Data and Power Generation: The study mentions typhoons and their effects on wind speeds, especially during summer, but it does not delve into how extreme wind events (like typhoons) specifically influence the energy conversion potential or operational challenges of wind turbines. There seems to be a gap in understanding the impact of extreme, short-duration wind events on the overall power generation efficiency and turbine lifespan.

Questions

1. Hourly Wind Speed Evaluation (Section 4.1 - a)

• How do you account for sudden, short-term fluctuations in wind speed when evaluating hourly wind speeds, especially during transition periods like 11 a.m. to 12 p.m.?

• Could you explain how the correlation between energy demand and wind speed might vary over different geographic areas, considering factors like population density and regional energy needs?

2. Monthly Wind Speed Variations (Section 4.1 - b)

• How significant are the variations in wind speed during the spring, and how do these low-speed months affect the feasibility of wind power generation in regions like Fujian?

• What mitigation strategies can be implemented to address periods with lower wind speeds (such as in May) to ensure consistent energy production?

3. Weibull Distribution (Section 4.1 - c)

• Can you provide more insight into why the scale parameter increases with height? Is this a consistent trend across different geographical regions or specific to the stations you analyzed?

• How do the differences in the shape and scale parameters at different heights influence the choice of turbine specifications for these regions?

4. Wind Power and Energy Density (Section 4.2)

• What factors contribute to the discrepancy between wind speed and energy density, and how might these affect long-term wind energy projections in Fujian's offshore regions?

• Could the wind roses data be used to better understand the distribution of energy production across different directional sectors? How might this be integrated into wind farm design?

5. Turbulence Intensity (Section 4.3)

• How do variations in turbulence intensity between the different classes (A, B, C) affect turbine performance over time, especially during typhoon season?

• Does the relatively low turbulence intensity found in the data (below 10%) indicate that the region is suitable for offshore wind farms, or does this suggest that special turbine designs might be required?

6. Wind Shears (Section 4.4)

• Why does the wind shear exponent (α) exhibit such seasonal variability, and how does this affect turbine placement and operational performance, especially for offshore turbines in Fujian?

• Given that the wind shear exponent is lower during the day compared to nighttime, how does this impact energy generation efficiency across different times of day?

7. Extreme Wind Speed (Section 4.5)

• Could you explain how the extreme wind speed data might influence turbine maintenance schedules and lifetime predictions, especially considering the recurrence of typhoons?

• How reliable are the MCP forecasts for extreme wind speeds over a 30-year period, and what is the associated uncertainty in long-term wind power forecasting?

8. General Summary & Recommendations

• Based on the extreme wind speeds observed in Xiapu, would you recommend any adjustments to turbine specifications for other coastal regions in China, particularly those prone to typhoons?

• How does the variability in wind shear and turbulence intensity affect the scalability of offshore wind projects in the region, particularly when considering the long-term operation of large-scale wind farms?

Reviewer #3: General Comment:

The manuscript presents a promising study; however, several areas need improvement to enhance its clarity, readability, and scientific contribution. The authors should address issues related to language quality, novelty presentation, and proper referencing. Additionally, improving the figures, expanding the literature review, and providing more detail in the conclusions section would strengthen the manuscript significantly. Below are more specific comments to guide the authors in revising the paper.

1. Language Quality:

o The manuscript would benefit from a thorough review of its English language usage to improve clarity and readability. It is recommended that the authors address grammatical and syntactic errors throughout the text to enhance the quality of the writing.

2. Novelty and Original Contribution:

o The novelty of the research is not clearly highlighted. The authors should explicitly articulate the unique contributions of their work to distinguish it from existing studies.

3. Nomenclature Section:

o To improve the manuscript's readability, the authors are encouraged to include a nomenclature section at the beginning. This should contain all variables, acronyms, indices, and constants used in the manuscript, allowing readers to refer easily to these definitions.

4. Introduction and Literature Review:

o The Introduction would benefit from a more thorough review of relevant literature. A comprehensive manuscript in this domain typically references a minimum of 40 studies. Expanding the literature review will also provide a stronger foundation for the study's relevance and innovation.

5. Recommended References:

o The following references are highly relevant to the subject and should be incorporated to enhance the background and context of the study:

Detection of internal fault in differential transformer protection based on fuzzy method. International Journal of Physical Sciences, 6(26), pp.6150-6158, 2011.

Distributed generation management in smart grid with the participation of electric vehicles with respect to the vehicle owners’ opinion by using the imperialist competitive algorithm. Sustainability, 14(8), p.4770, 2022.

Optimum operation management of microgrids with cost and environment pollution reduction approach considering uncertainty using multi‐objective NSGAII algorithm. IET Renewable Power Generation, 2022.

Economic dispatch optimization considering operation cost and environmental constraints using the HBMO method. Energy Reports, 10, pp.1718-1725, 2023.

Unit commitment for power generation systems based on prices in smart grid environment considering uncertainty. Sustainability, 13(18), p.10219, 2021.

Locating and sizing of capacitor banks and multiple DGs in distribution system to improve reliability indexes and reduce loss using ABC algorithm. Bulletin of Electrical Engineering and Informatics, 10(2), pp.559-568, 2021.

Comparison of SVC and STATCOM in static voltage stability margin enhancement. International Journal of Electrical and Computer Engineering, 3(2), pp.297-302, 2009.

Optimal management of energy storage systems for peak shaving in a smart grid. Computers, Materials and Continua, 75(2), pp.3317-3337, 2023.

Optimal design of solar–wind hybrid system-connected to the network with cost-saving approach and improved network reliability index. SN Applied Sciences, 1(12), p.1742, 2019.

6. Figures Quality:

o Figures in the manuscript should be provided in higher resolution to improve clarity. Each figure should also be thoroughly described within the text to ensure that it contributes effectively to the manuscript's presentation.

7. Future Work and Limitations:

o A brief discussion of potential limitations and directions for future research should be included in the Conclusions and Recommendations section. This will provide a more comprehensive view of the study’s scope and areas for future exploration.

Reviewer #4: The authors conducted a good structure and a comprehensive assessment of offshore wind resource at a site near Fujian China. I could say that I enjoyed reading this work entitled " Wind resource assessment Offshore Fujian using 30-year wind estimates". However, there are some points to be addressed.

1- The research gaps in prior studies are not highlighted clearly. It is better to mention the research gap and how this research addresses those in a novel approach. Moreover, the novelty of the work must be clearly addressed and discussed, compare your research with existing research findings and highlight novelty, (compare your work with existing research findings and highlight novelty)

2- The study uses data from two meteorological stations in the Fujian region. What is the novelty in choosing this particular location, and how does the study contribute to the wider knowledge of offshore wind resources in China or globally?

3- The study utilizes the MCP method to predict wind conditions for a 30-year span. Please clarify how robust the MCP technique is in accurately representing long-term wind behavior, especially considering the variability in seasonal and inter-annual patterns of wind speeds? and how the results can be validated.

4- In the conclusion, please discuss the limitations of their study, such as the use of the MCP method and seasonal variability in wind data. Also, highlight key findings, like predicted wind speeds and energy density values, and suggest potential directions for future research in offshore wind resource assessments and how it can be integrated to the energy sector..

Reviewer #5: This study presents a comprehensive assessment of offshore wind resources along the Fujian coastal zone. The authors deployed anemometric equipment at two sites collecting one year of wind measurements including speed, direction, and turbulence intensity at 100 meters above mean sea level. To address the limitation of single-year measurement duration, a MCP methodology was employed to estimate 30-year wind conditions using atmospheric reanalysis data. The investigation evaluates the suitability of different wind turbine classes for this offshore site based on extreme wind speed and turbulence intensity estimates.

While the paper represents a meaningful observational effort with publishable data, several improvements could enhance its scientific rigor:

1. A more thorough examination of vertical profile characteristics across measurement heights and seasonal variations in wind patterns would strengthen the resource assessment.

2. Additional analysis regarding the applicability and accuracy of the MCP approach in this marine environment would improve confidence in the long-term estimates.

3. The turbine suitability analysis could be augmented by addressing local environmental challenges such as typhoon impacts, marine meteorological conditions, and their implications for turbine class selection.

4. The conclusion should better highlight the study's novel contributions to offshore wind assessment methodologies and regional renewable energy planning.

5. Figure Clarifications:

a. Units specification for "frequency" in Figures 4-5

b. Please improve visualization of scattered data in Figure 7

c. Correction of duplicate "100m" entry in Table 4 (Xiapu station parameters)

6. PLOS authors have the option to publish the peer review history of their article (what does this mean? ). If published, this will include your full peer review and any attached files.

**Do you want your identity to be public for this peer review?** For information about this choice, including consent withdrawal, please see our Privacy Policy .

Reviewer #1: No

Reviewer #2: No

Reviewer #3: No

Reviewer #4: No

Reviewer #5: No

---

## [Author Response · Author response to Decision Letter 1]

25 Apr 2025

Reply to Reviewer #1

First of all, the authors are very grateful to the reviewer #1 to make the very constructive comments on our manuscript. We have realized that some places in our manuscript were unclear or improperly presented, not limited to the places mentioned below. We have taken full consideration of these comments and submitted a revised manuscript. In this point-by-point response, we reproduced the comments (black font), gave our responses (red font), and highlighted the revisions in new version of the paper in a red color.

We make the following major revision and reply to comment by the reviewer #1.

Reviewer #1: Below are my comments regarding the manuscript:

- Abstract: The abstract requires grammatical revisions and careful proofreading for spelling errors. A clearer articulation of the main findings would enhance its effectiveness.

Reply: Thank you for your good suggestion! The abstract has been rewritten carefully, and present the main findings of the study.

- Introduction: The introduction section needs a comprehensive revision. Currently, it lacks focus and should be condensed to present the key points more succinctly.

Reply: Good suggestion, the introduction has been reorganized and condensed. The revised introduction focus on current status and issues of offshore wind resource assessment in China, highlighting the significance of this study.

- References: The manuscript would benefit from the inclusion of more recent references. For instance, the following article discusses the statistical properties (mean, variance, and standard deviation) of wind turbine blades and could provide valuable insights: https://doi.org/10.1177/08927057231154427.

Reply: The effects of the size and location of magnetorheological fluid segments on random vibration characteristics on turbine blades are investigated in the paper. In our work, we emphasize the environment characteristic, which is useful for design of turbine blades.

- Conclusion: The conclusion should be enhanced to provide a clearer overview of the scope of the work and the key achievements. A narrative format rather than a bullet-point list would improve clarity and coherence.

Reply: Thank you for your good suggestion! We have revised the “Section 5 summary and conclusions”. Summarized the scope of the work, and highlighted the key achievements.

- Formatting of Equations: Consistency in the font used for all mathematical formulations is necessary. Please ensure that all formulas are uniformly formatted and revised.

Reply: Thank you! We have revised the formatting of Equations according your suggestion.

- Overall Framework: The overall structure of the article requires significant revision. A more logical flow of information would improve readability and comprehension.

Reply: We have reorganized the structure of article. Section 2 provides an overview of the study and observation data. Section 3 presents detailed analysis method, including wind speed distribution model, most probable wind speed, air density calculation, wind energy density, wind shears, turbulence intensity and Measure–Correlate–Predict method. The results concerning the high frequency variability of wind and wind power and extreme return winds at a height of 100 meters is analyzed in Section 4. Finally, Section 5 provides a summary and conclusions.

- Writing Quality: Please pay attention to the overall writing quality and grammar throughout the manuscript. Enhancing these aspects will strengthen the presentation of your arguments.

Reply: Thank you! We carefully revised the manuscript, and look forward to meet the publish standards.

- Figure and Table Captions: All captions for figures and tables should be carefully revised for clarity and completeness. Ensure they accurately reflect the content.

Reply: Thank you! We have revised all captions for figures and tables according your good suggestion.

- Figure 10: The explanation accompanying Figure 10 should be expanded. Currently, it is not sufficiently clear, and a more detailed description would help convey the intended message effectively.

Reply: Scattered data (Figure 10) in original manuscript is used to visualize the relation between observed wind speed and reference wind speed from ERA5. In the revised version, the figure is omitted and only the result is present in the Section 4.6, considering too many figures involving in the manuscript.

Reply to Reviewer #2

First of all, the authors are very grateful to the reviewer #2 to make the very constructive comments on our manuscript. We have realized that some places in our manuscript were unclear or improperly presented, not limited to the places mentioned below. We have taken full consideration of these comments and submitted a revised manuscript. In this point-by-point response, we reproduced the comments (black font), gave our responses (red font), and highlighted the revisions in new version of the paper in a red color.

We make the following major revision and reply to comment by the reviewer #2.

Reviewer #2: Comments for authors:

Key Areas Needing Revision:

Main Gaps:

1. Seasonal and Temporal Variability in Wind Energy Density: While the paper provides a thorough analysis of wind speed and its relationship with energy density, it does not fully address how the temporal and seasonal fluctuations in wind energy density might impact the long-term feasibility and efficiency of offshore wind farms. The variations in seasonal wind energy density suggest that energy production may fluctuate more significantly than wind speed alone indicates, yet the study lacks a detailed analysis of these fluctuations in the context of wind farm operations over multiple years.

Reply: Thank you! We have added the analysis of temporal and seasonal fluctuations in wind energy density.

“The notable monthly variation highlights the necessity of differentiating between various months of the year when evaluating or designing a wind power project. Spring in Fujian exhibits pronounced wind speed fluctuations with larger TI due to seasonal transitions and regional meteorological dynamics. For instance, wind speeds can drop below 6 m/s during May, while episodic gusts from monsoons or typhoon precursors may exceed 15 m/s. Furter statistical analysis reveals that calm period (wind speed < 3 m/s) account for ~30% of spring days in Fujian, significantly reducing the effective power generation window. Wind turbines typically require a minimum cut-in speed of 3–4 m/s. During low-speed months, Fujian’s capacity factor (CF) drops to 15–20%, compared to 35–40% in high-wind seasons (summer and autumn). This seasonality necessitates overcapacity design or hybrid energy systems to stabilize supply, such as integrating solar or tidal energy to compensate for wind intermittency.”

“To ensure consistent energy production with lower wind speed period (e.g., in May), Some mitigation strategies should be implemented: 1) Technical optimization of wind turbines, deploy turbines with larger rotor diameters (e.g., 140–150m) and lower-rated capacities to capture low-speed wind energy more efficiently. For example, Goldwind's GW115/2.0 model increased annual output by 10.6% at 100m hub heights compared to 85 m towers in low-wind regions. Use hybrid steel-concrete or full-steel flexible towers (120–140m) to exploit higher wind speeds at elevated heights, consider the positive wind shear coefficients in the area. Optimize power tracking across low-to-medium wind speeds (2.5–5 m/s) through permanent magnet generators, achieving near-optimal power coefficient throughout the entire operational range. 2) Energy storage integration, deploy battery storage systems to store excess energy during high-wind periods and discharge during low-wind intervals, smoothing output fluctuations. Implement pumped hydro or hydrogen electrolysis systems for multi-day energy reserves. 3) Constructing multi-energy complementary systems, integrate photovoltaic panels to compensate for wind intermittency. For instance, solar generation typically peaks in summer, offsetting reduced wind output.”

2. Effect of Typhoons on Wind Data and Power Generation: The study mentions typhoons and their effects on wind speeds, especially during summer, but it does not delve into how extreme wind events (like typhoons) specifically influence the energy conversion potential or operational challenges of wind turbines. There seems to be a gap in understanding the impact of extreme, short-duration wind events on the overall power generation efficiency and turbine lifespan.

Reply: In the revised manuscript, we have added discussion of extreme wind events, which influence the energy conversion potential and operational challenges of wind turbines.

“Extreme wind speeds during typhoons impose non-linear mechanical stresses on turbine components (e.g., blades, gearboxes, and towers). Typhoon-induced gusts exceeding 25 m/s can cause blade bending moments exceeding design limits, accelerating microcrack propagation. High turbulence intensity during typhoons increases gearbox vibration amplitudes by 30–50%, shortening lubrication cycles. These necessitates shorter inspection intervals (e.g., post-typhoon structural checks) and proactive replacement of fatigue-prone parts.”

Questions

1. Hourly Wind Speed Evaluation (Section 4.1 - a)

• How do you account for sudden, short-term fluctuations in wind speed when evaluating hourly wind speeds, especially during transition periods like 11 a.m. to 12 p.m.?

Reply: We added the description hourly wind speed variation and mechanism analysis in the revised manuscript:

“Hourly wind speeds indicate that near-shore wind speeds in Fujian typically exhibit a bimodal variation. During morning surge (02:00-12:00), wind speeds increase by 15-20% compared to nighttime averages. During afternoon peak (14:00-20:00), maximum wind speeds occur, often exceeding 9 m/s above 70m height. Speeds gradually decrease after 00:00, with minimum values occurring before dawn. Wind speed differences between daytime peaks and nighttime lows average 2-3 m/s, reaching 4-5 m/s during winter monsoon periods. This diurnal signature is most pronounced in summer when thermal forcing dominates over synoptic systems. Such wind modal is related to Land-Sea thermal forcing. Solar heating intensifies land-sea temperature contrast during daytime, generating sea breezes that amplify wind speeds. The coastal terrain funnels these thermally-driven flows, creating acceleration zones. At night, reversed land breezes interact with background monsoon winds, causing partial speed reduction. Secondly, the Taiwan Strait's "wind tunnel" effect mechanically accelerates northwest monsoon winds. This effect peaks when solar heating strengthens pressure gradients across the strait. In addition, daytime convective mixing transports momentum from upper levels downward, enhancing surface winds. Stable nighttime atmospheric stratification inhibits this process.”

• Could you explain how the correlation between energy demand and wind speed might vary over different geographic areas, considering factors like population density and regional energy needs?

Reply: We have added the explanation of the correlation between energy demand and wind speed in the revised manuscript, thank you for your suggestion!

“The correlation between energy demand and wind speed exhibits significant geographic variability, influenced by three key factors: 1) Wind resource distribution, China's southeastern seas experience stronger and more consistent wind speeds, enabling higher wind energy penetration. In contrast, Bohai Sea and Yellow Sea with low wind potential show weaker correlations due to limited wind-driven energy supply. 2)Energy demand primarily stems from industrial/commercial activities in high-density urban zones (e.g., coastal cities of China). Decentralized wind farms often align energy production with local demand in low-density rural areas. 3) Advanced transmission networks dilute local wind-speed-demand correlations by integrating diverse energy sources for well grid-connected regions. This geographic heterogeneity underscores the need for region-specific wind-energy integration strategies, particularly in developing hybrid systems that balance variable wind resources with population-driven demand profiles.”

2. Monthly Wind Speed Variations (Section 4.1 - b)

• How significant are the variations in wind speed during the spring, and how do these low-speed months affect the feasibility of wind power generation in regions like Fujian?

Reply: We have added the analysis above the monthly wind speed variations in Section 4.1-b as follows: “The notable monthly variation highlights the necessity of differentiating between various months of the year when evaluating or designing a wind power project. Spring in Fujian exhibits pronounced wind speed fluctuations with larger TI due to seasonal transitions and regional meteorological dynamics. For instance, wind speeds can drop below 6 m/s during May, while episodic gusts from monsoons or typhoon precursors may exceed 15 m/s. Furter statistical analysis reveals that calm period (wind speed < 3 m/s) account for ~30% of spring days in Fujian, significantly reducing the effective power generation window. Wind turbines typically require a minimum cut-in speed of 3–4 m/s. During low-speed months, Fujian’s capacity factor (CF) drops to 15–20%, compared to 35–40% in high-wind seasons (summer and autumn). This seasonality necessitates overcapacity design or hybrid energy systems to stabilize supply, such as integrating solar or tidal energy to compensate for wind intermittency.”

• What mitigation strategies can be implemented to address periods with lower wind speeds (such as in May) to ensure consistent energy production?

Reply: We have added the analysis above the monthly wind speed variations in Section 4.1-b as follows:

“To ensure consistent energy production with lower wind speed period (e.g., in May), Some mitigation strategies should be implemented: 1) Technical optimization of wind turbines, deploy turbines with larger rotor diameters (e.g., 140–150m) and lower-rated capacities to capture low-speed wind energy more efficiently. For example, Goldwind's GW115/2.0 model increased annual output by 10.6% at 100m hub heights compared to 85 m towers in low-wind regions. Use hybrid steel-concrete or full-steel flexible towers (120–140m) to exploit higher wind speeds at elevated heights, consider the positive wind shear coefficients in the area. Optimize power tracking across low-to-medium wind speeds (2.5–5 m/s) through permanent magnet generators, achieving near-optimal power coefficient throughout the entire operational range. 2) Energy storage integration, deploy battery storage systems to store excess energy during high-wind periods and discharge during low-wind intervals, smoothing output fluctuations. Implement pumped hydro or hydrogen electrolysis systems for multi-day energy reserves. 3) Constructing multi-energy complementary systems, integrate photovoltaic panels to compensate for wind intermittency. For instance, solar generation typically peaks in summer, offsetting reduced wind output.”

3. Weibull Distribution (Section 4.1 - c)

• Can you provide more insight into why the scale parameter increases with height? Is this a consistent trend across different geographical regions or specific to the stations you analyzed?

Reply: We have added the analysis for scale parameter variation in Section 4.1-c as follows:

“The scale parameter of the Weibull distribution reflects the characteristic wind speed magnitude. Its increase with height primarily stems two seasons. Firstly, Wind shear diminishes with altitude due to decreased ground roughness, allowing higher wind speeds. Secondly, the vertical wind profile typically follows a power-law relationship (Eq.13, Figs.11 and 12), causing scale parameter to scale proportionally with height. Although the general trend of parameter increasing with height holds globally, its magnitude varies regionally due to different terrain and atmospheric stability condition.”

• How do the differences in the shape and scale parameters at different heights influence the choice of turbine specifications for these region

---

## [Decision Letter · Decision Letter 1]

Wind resource assessment Offshore Fujian using 30-year wind estimates

PONE-D-24-45330R1

Dear Dr. Guanghong Liao,

We’re pleased to inform you that your manuscript has been judged scientifically suitable for publication and will be formally accepted for publication once it meets all outstanding technical requirements.

Kind regards,

Soheil Mohtaram

Academic Editor

PLOS ONE

Additional Editor Comments:

After a careful and thorough evaluation of all the reviewers’ comments, I made the decision to accept the manuscript based on the following rationale:

Three out of five reviewers recommended acceptance, and their comments indicated that the manuscript meets the publication criteria of PLOS One. The fourth reviewer, while not explicitly recommending acceptance, provided comments that were generally constructive and did not raise any critical issues that would preclude publication. I reviewed this report in detail and found the feedback to be satisfactorily addressed by the authors. The only reviewer who recommended rejection raised concerns that, although respected, appeared either overly critical or subjective, and did not, in my assessment, outweigh the comprehensive and positive evaluations from the majority. In my role as Academic Editor, I believe it is essential to consider the collective input of all reviewers, and in this case, the majority clearly supported the manuscript’s publication. Furthermore, based on my own reading and evaluation, I am personally confident that the manuscript demonstrates sufficient novelty, methodological soundness, and relevance to the journal's scope. I am therefore satisfied that, in its current form, the manuscript meets all the publication criteria of PLOS One.

Reviewers' comments:

Review Comments to the Author

Reviewer #1: 

Reviewer #3: Comments:

Lack of Clear Novelty: The methods and findings appear to replicate existing studies, and the manuscript does not clearly demonstrate what new contributions it offers to the field.

Insufficient Methodological Detail: The manuscript lacks adequate explanation of the data sources, analytical approaches, and validation procedures, making it difficult to assess the scientific rigor.

Issues with Language and Structure: The manuscript contains numerous grammatical errors and structural weaknesses that detract from its clarity and professionalism.

Weak Interpretation of Results: The analysis and discussion of the results are rather superficial and do not effectively highlight their implications or practical significance.

Reviewer #4: Thanks for authors responses. All comments are well addressed in the revised manuscript and it is suitable for publication.

Reviewer #5: The reviewer appreciates that all of my concerns have been considered by the authors and agrees the publication of the paper in its present form.

---

## [Editor Report · Acceptance letter]

PONE-D-24-45330R1

PLOS ONE

Dear Dr. Liao,

I'm pleased to inform you that your manuscript has been deemed suitable for publication in PLOS ONE. Congratulations! Your manuscript is now being handed over to our production team.

Kind regards,

on behalf of

Dr. Soheil Mohtaram

Academic Editor

PLOS ONE